# Bandit Theory and Thompson Sampling-Guided Directed Evolution for Sequence Optimization

Hui Yuan[1], Chengzhuo Ni[2], Huazheng Wang[3], Xuezhou Zhang[4], Le Cong[5], Csaba Szepesvári[6,7], and Mengdi Wang[7,8]

[1,2,4,8]Department of Electrical and Computer Engineering, Princeton University
[3]School of Electrical Engineering and Computer Science, Oregon State University
[5]Department of Pathology and Department of Genetics, Stanford University
[6]Department of Computing Science, University of Alberta
[7]DeepMind [*†]

## Abstract

Directed Evolution (DE), a landmark wet-lab method originated in 1960s, enables discovery of novel protein designs via evolving a population of candidate sequences. Recent advances in biotechnology has made it possible to collect high-throughput data, allowing the use of machine learning to map out a protein's sequence-to-function relation. There is a growing interest in machine learning-assisted DE for accelerating protein optimization. Yet the theoretical understanding of DE, as well as the use of machine learning in DE, remains limited. In this paper, we connect DE with the bandit learning theory and make a first attempt to study regret minimization in DE. We propose a Thompson Sampling-guided Directed Evolution (TS-DE) framework for sequence optimization, where the sequence-to-function mapping is unknown and querying a single value is subject to costly and noisy measurements. TS-DE updates a posterior of the function based on collected measurements. It uses a posterior-sampled function estimate to guide the crossover recombination and mutation steps in DE. In the case of a linear model, we show that TS-DE enjoys a Bayesian regret of order $\widetilde{O}(d^2\sqrt{MT})$, where $d$ is feature dimension, $M$ is population size and $T$ is number of rounds. This regret bound is nearly optimal, confirming that bandit learning can provably accelerate DE. It may have implications for more general sequence optimization and evolutionary algorithms.

## 1 Introduction

Protein engineering means to design a nucleic acids sequence for maximizing a utility function that measures certain fitness or biochemical/enzymatic properties, i.e., stability, binding affinity, or catalytic activity. Due to the combinatorial sequence space and lack of knowledge about the sequence-to-function map, engineering and identifying optimal protein designs were a quite daunting task. It is only until recently that synthesis of nucleic acid sequences and measurement of protein

---

[*]Authors' emails are: {huiyuan, cn10, xz7392, mengdiw}@princeton.edu, huazheng.wang@oregonstate.edu, congle@stanford.edu, szepesva@ualberta.ca.

[†]Mengdi Wang acknowledges support by NSF grants DMS-1953686, IIS-2107304, CMMI-1653435, and ONR grant 1006977. Le Cong acknowledges support by NIH grant R35-HG011316, and Donald and Delia Baxter Foundation grant, and NSF grant DMS-1953686. Csaba Szepesvári gratefully acknowledges the funding from Natural Sciences and Engineering Research Council (NSERC) of Canada, "Design.R AI-assisted CPS Design" (DARPA) project and the Canada CIFAR AI Chairs Program for Amii.

function became reasonably scalable [41, 57], allowing rational optimization or directed evolution of protein designs. Nonetheless, because of the complex landscape of protein functions and the bottleneck of wet-lab experimentation, this remains a very difficult problem.

Directed evolution (DE), one of the top molecular technology breakthroughs in the past century, demonstrates human's ability to engineer proteins at will. DE is a method for exploring new protein designs with properties of interest and maximal utility, by mimicking the natural evolution. It works by artificially evolving a population of variants, via mutation and recombination, while constantly selecting high-potential variants [8, 9, 33, 25, 49, 41]. The development of directed evolution methods was honored in 2018 with the awarding of the Nobel Prize in Chemistry to Frances Arnold for evolution of enzymes, and George Smith and Gregory Winter for phage display [4, 47, 53]. See Figure 1.1 for illustrations of mutation and crossover recombination.

DE practitioners' major considerations center on cost and data quality. First, the ability to synthesize and mutate new biological sequences have been exponentially improved thanks to synthetic chemistry advances. Second, given a population of sequences $S$, selecting and identifying the set of optimal sequences is straightforward, using low-cost parallel sequencing which works well with pooled selection assays. Third, using pooled measurement to evaluate the average value of protein function (mean fitness) over a population $S$ is generally easy, as such bulk measurements is low-cost and high-quality. Finally, querying $f(x)$ for a given $x$ is often expensive and time-consuming, and the cost adds up quickly if many queries are needed. It can be desirable to perform this procedure in small-scale batches to optimize time and resource consumption.

Such difficulties have motivated scientists to apply machine learning approaches to accelerate DE, beginning with Fox et al. [16] and followed

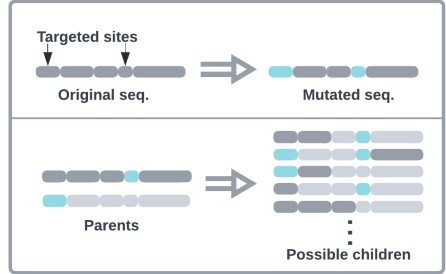

Figure 1.1: Illustration of mutation and crossover recombination. Mutating a sequence means to replace a targeted or random entry (site) by a random or designated value. Recombination involves two or multiple sequences. For example, parent sequences can crossover, exchange subsequences and generate children.

by many. Recent development of directed evolution have increasingly utilized *in silico* exploration and machine learning beyond experimental approaches [57, 15, 12, 45, 18, 50, 46]. While these attempts have proved to be successful in simulation and sometimes in real experiments, little is known about the statistical theory of DE.

In this paper, a primary objective is to bridge the directed evolution process with bandit learning theory. In particular, we want to express machine learning-assisted DE as a bandit optimization process, with a theoretical justification. Further, we aim to understand how a machine learning model, as simple as linear, can accelerate DE and reduce the overall cost of evaluation. Specifically, we propose a Bayesian bandit model for DE, namely the Thompson Sampling-guided Directed Evolution framework, which combines posterior model sampling with directed mutation and recombination. The theoretical analysis shows that the crossover selection mimics an optimization iteration, and the optimization progress is proportional to a level of population diversity. In the case of the linear model, we establish a Bayesian regret bound $\widetilde{O}(d^2\sqrt{MT})$[3] that depends polynomially on feature dimension $d$[4], and optimally in batch size $M$ and time steps $T$. We finally harmonize our theoretical analysis with a set of simulation and real-world experiment.

**Important Remark** The scope of this work is to provide a simplified mathematical model and basic theoretical understanding of an evolutionary-based process that is common in directed evolution. We emphasize that our framework is a theoretical simplification, assuming linear objective over a hy-

---

[3] $\widetilde{O}(\cdot)$ ignores the logarithmic terms.

[4] Our TS-DS regret has two extra $\sqrt{d}$ factors compared to the classic result that the optimal regret for linear bandits is of $\widetilde{O}(d\sqrt{MT})$. One $\sqrt{d}$ comes from our problem setting where the $l_2$ norm of each action is $O(\sqrt{d})$. Another factor of $\sqrt{d}$ is due to the evolutionary nature of DE. See the remark in §5.2 for more details.

percube. In real-world experimental systems, one needs to consider prior knowledge about the system and applying any machine learning method would require careful calibration and customization.

## 2 Related work

From a theoretical perspective, our analysis is related to the literature on evolutionary algorithms and linear bandits.

**Evolutionary algorithm.** The success of DE motivated a large body of works on evolutionary algorithms for optimization. Evolutionary algorithm (EA) [6] is a large class of randomized optimization algorithms, based on the heuristic of mimicking natural evolution. Despite many variants, a typical EA usually maintains a population of solutions and improves the solutions by alternating between reproduction step which produces new offspring solutions, and selection step where solutions are evaluated by the objective function and only the good ones are saved to the next round. Theoretical understandings of EA are focusing on specific EAs, among which the most well-studied setting is $(1+1)$-EA, with parent population size and offspring population size are both 1 to optimize linear objective function on the Boolean space $\{0,1\}^d$, see [14, 24, 27, 28, 35, 55]. EA analysis focuses on optimization and reducing the running time instead of minimizing total regret as in bandit theory. There are other results on population based EAs, such as $(1+\lambda)$-EA [11, 19], $(\mu+1)$-EA [54] and the most general $(\mu+\lambda)$-EA, where $\mu$ and $\lambda$ represent the parent population size and the offspring population size respectively. However, this group of works only adopted mutation. The understanding of the role played by recombination in evolutionary algorithms was left as blank in the $(\mu+\lambda)$-EA framework, while our paper provides a population-based regret minimization analysis with both mutation and recombination.

There are a few works [30, 29, 51, 32] studying EAs with recombination (which are also called genetic algorithms (GAs)). However, their algorithms and analysis are tailored to artificial test objectives and the results are not able to generalize even to linear objectives. Recently, the running time analysis of some natural EAs with recombination has been conducted [39, 40], but still their results are constrained under specific objectives such as ONEMAX and JUMP. We refer readers to the book by [59] for a more comprehensive review of EA.

**Linear bandits.** Bandit is a powerful framework formulating the sequential decision making process under uncertainties. Under this framework, linear bandits is a central and fruitful branch where in each round a learner makes her decision and receives a noisy reward with its mean value modelled by a linear function of the decision, aiming to maximize her total reward (or minimize total regret equivalently) over multiple rounds [5, 36, 1], with extension to sparse linear bandits [23] and linear MDP [58]. In the same spirit, the process evolving a population of genetic sequences to maximize a linear utility over the evolution trajectory, while getting access to noisy utility values through evaluating sequences along the way, can be mathematically formulated from the perspective of linear bandits. One of the main solutions in linear bandits is the upper confidence bound-based (UCB) strategy represented by LinUCB [36], where the learner makes decision according to upper confidence bounds of the estimated reward and the accumulated regret is proven to be $\widetilde{O}\left(d\sqrt{T}\right)$. A similar strategy is optimism in the face of uncertainty (OFU) principle in Abbasi-Yadkori et al. [1]. The other approach is the Thompson Sampling (TS) strategy, which randomizes actions on the basis of their probabilities to be optimal. Russo and Van Roy [43] proved the Bayesian regret of TS algorithm is also of order $\widetilde{O}\left(d\sqrt{T}\right)$. And there are more results on the regret of TS(-like) algorithms solving linear bandits in the frequentist view [3, 2, 21, 3]. We also refer readers to the book by [34] for a delicate review of bandit theory.

**Non-evolutionary methods for protein sequence design.** Though our framework applies to an evolution-based DE process, there exist many other methods that are not evolution-based. Protein engineering is a rich field and it is not restricted to methods that are based on mutagenesis and recombination. Protein sequence engineering constantly evolves as new bio-technologies keep emerging. For example, new biotechnology makes it possible to synthesize specific variants and operate on the combinatorial space likewise with high-throughput method, and this allows directly applying a Gaussian process bandit algorithm [42]. See [57] for a high-level survey of this active

area of research, and see [17, 7] for more examples. This active and exciting field brings many new opportunities for machine learning.

**Remark.** It is important to note that our problem is *not* a multi-armed bandit problem. In bandits, one can choose actions freely from the full action set. However, in biological experiments, it is expensive to synthesize a new protein design sequence out of thin air. Instead, mutation and recombination are used to generate new designs easily at a low cost. Thus our algorithm can only guide the selection step in the DE process. Its regret is not directly comparable with the regret of multi-arm bandits. To the best of our knowledge, this is the first work that studies the bandit theory and regret bound of mutation and recombination-enabled DE.

## 3 Bandit model for directed evolution

### 3.1 Process overview

We illustrate the Thompson Sampling-guided Directed Evolution (TS-DE) process in Figure 3.1. A population $S_t$ at time $t$ consists of $M$ candidate sequences. It evolves via mutation, crossover recombination, selection, and function evaluation to the next generation $S_{t+1}$. The mutation and crossover selection are guided using a learnt function $f_{\tilde{\theta}_t}$, in order to filter out unwanted candidates and keep only a small batch for costly evaluation. Collected data are fed into a Thompson Sampling module for posterior update of $f_{\tilde{\theta}_t}$. Full details of the mutation, crossover selection, and Thompson Sampling modules will be given in Section 4.

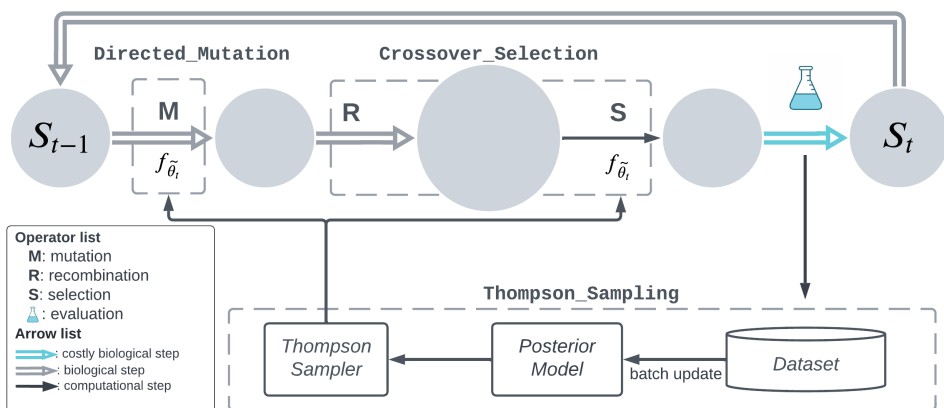

Figure 3.1: **Thompson sampling-guided directed evolution**

### 3.2 Motif feature, utility model, recombination and mutation operators

A genetic sequence comprises of functional motifs, i.e., functional subsequences that may encode particular features of protein, also known as protein motifs [37, 48, 10]. Such genetic motifs are known to be "evolutionarily conserved", in the sense that they tend to evolve as units, under mutation and recombination.

Suppose a genetic sequence $seq$ is made up of $d$ genetic motifs, given by $seq = (seq_{(1)}, seq_{(2)}, \cdots, seq_{(d)})$. Machine learning models for protein utility prediction are often based on motif features [56, 10, 38]. Let $\mathcal{X}$ be the space of genetic sequences of interest. We assume that a binary motif feature map is given, defined as follows.

**Definition 3.1** (Binary Motif Feature Embedding). Let $\phi$ be the genetic motif feature map given by:

$$\phi : \mathcal{X} \to \{0, 1\}^d, \quad \phi(seq) := (\phi_1(seq_{(1)}), \cdots, \phi_d(seq_{(d)})) \tag{3.1}$$

such that at each dimension $i$, $\phi_i(seq_{(i)})$ is a binary feature of motif $seq_{(i)}$.

The binary motif feature provides a minimalist abstraction for evolutionary processes where $0, 1$ correspond to favorable and nonfavorable directions, respectively, for each motif. Theoretical analysis for evolutionary optimization algorithms made the same assumption and viewed binary sequence optimization as a fundamental problem [14, 24, 27, 28, 35, 55].

Since a protein function is largely determined by its motif, it is common to model the protein utility $f : \mathcal{X} \to \mathbb{R}$ as a function of motif features, i.e., $f(seq) := f_{\theta^\star}(x), x = \phi(seq), \forall seq \in \mathcal{X}$, under a parameterization by $\theta^\star$ [16, 57, 45, 18].

In this work, we study the most elementary Bayesian linear model, where $f$ is a linear model parameterized by $\theta^*$ with a Gaussian prior, given as follows.

**Assumption 3.2.** (Linear Bayesian Utility Model) Assume the utility $f_{\theta^\star}$ is a linear function parameterized by $\theta^\star \in \mathbb{R}^d$, which is sampled from a Gaussian prior, i.e.

$$f_{\theta^\star}(x) = \langle \theta^\star, x \rangle, \qquad \theta^\star \sim \mathcal{N}(\mathbf{0}, \lambda^{-1}\mathbf{I}), \quad \lambda > 0. \tag{3.2}$$

Since motifs tend to mutate and recombine with one another in units, it is often sufficient to focus on recombination and mutation on the motif level, rather than on the entry level. Further, recombination that breaks a motif often result in insignificant low-fitness descendants. Therefore, it suffices to focus on motif-level directed evolution for simplicity of presentation and theory. For theoretical simplicity, we define recombination and mutation operators **on the motif level**:

**Definition 3.3** (Directed Mutation Operator). Let $x$ be the motif feature sequence, $\mathcal{I} \subset [d]$ be a collection of targeted sites and $\mu \in (0, 1)$ be a mutation rate. The mutation operator $\mathtt{Mut}(x, \mathcal{I}, \mu)$ generates a sequence $x'$ such that while for $\forall j \notin \mathcal{I}, x'_j = x_j$, for $\forall i \in \mathcal{I}, x'_i$ is independently induced to be

$$\begin{cases} x'_i \sim \mathrm{unif}(\{0, 1\}), & \text{w.p. } \mu, \\ x'_i = x_i, & \text{otherwise.} \end{cases} \tag{3.3}$$

**Definition 3.4** (Recombination Operator). Let $x, y$ be the motif features associated with two parental genetic sequences. The recombination operator $\mathtt{Rcb}(x, y)$ generates a child sequence $z$ such that $z_i$'s are independent and

$$z_i = \begin{cases} x_i & \text{w.p. } \frac{1}{2} \\ y_i & \text{w.p. } \frac{1}{2} \end{cases}, \ \forall i \in [d]. \tag{3.4}$$

We remark that Definitions 3.3, 3.4 are *mathematical simplifications* of their real-world counterparts. In real world, mutation and recombination can take various forms depending on the context. In our analysis, we define them in a minimalist-style to keep theory generalizable and interpretable.

## 3.3 Regret minimization problem formulation

Evaluating the protein function for a design sequence $x$ is a most costly and time-consuming step in protein engineering. In the DE process, we consider that regret is incurred only when sequences are evaluated. We also assume that each evaluation is subject to a Gaussian noise with known variance.

**Assumption 3.5.** (Noisy Feedback) Upon querying the utility of $x$, we get an independent noisy evaluation given by

$$u(x) \sim \mathcal{N}(f_{\theta^\star}(x), \sigma^2). \tag{3.5}$$

Our goal is to minimize the Bayesian regret, i.e., the cumulative sum of optimality gaps between evaluated sequences and the optimal.

**Definition 3.6** (Bayesian Regret). Denote by $f_{\theta^\star}(x^\star)$ the optimal utility value over $\mathcal{X}$, $\{x_{t,i}\}_{i=1}^M$ are the evaluated individuals in each iteration. Throughout $T$ iteration, the accumulated regret is defined as

$$\mathrm{BayesRGT}(T, M) = \mathbb{E}\left[\sum_{t=1}^T \sum_{i=1}^M (f_{\theta^\star}(x^\star) - f_{\theta^\star}(x_{t,i}))\right],$$

where $M$ is number of sequences selected for evaluation per timestep, and $\mathbb{E}$ is taken over the prior of $\theta^\star$ and all randomness in the DE process.

# 4 Thompson Sampling-guided directed evolution (TS-DE)

We restate our goal as to direct a population of genetic sequence to evolve towards higher utility value, until its population-average converges to the optimum $f_{\theta^\star}(x^\star)$. Our knowledge of $f$ is to be learned from noisy evaluations of selected sequences along the way. In this section, by integrating the biological technique - directed evolution - with Thompson Sampling, a Bayesian bandit method, we propose the Thompson Sampling-guided Directed Evolution algorithm (TS-DE) as shown in Alg 1, where in each round Thompson sampling gives an estimate of $\theta^\star$, based on which key operators of DE: mutation, recombination and selection are implemented.

## 4.1 Crossover-then-selection and directed mutation

Pairwise crossover is a most common type of recombination in natural evolution. Let $x, y$ be a random pair of parents, and let $z = \mathrm{Rcb}(x, y)$ be a child. If given a utility function $f$, we select $z$ only if the child performs better than the parents' average. Module 1 formulates this procedure.

---

**Module 1** $\mathtt{Crossover\_Selection}(f, S)$

---

1: **Inputs:** utility function $f(x) = \langle \theta, x \rangle$, a population of sequences $S$
2: **Initialization:** $S' \leftarrow \emptyset$
3: **while** $|S'| < |S|$ **do**
4:      Sample $x$ and $y$ from $S$ uniformly with replacement.
5:      **Recombination:** $z \leftarrow \mathrm{Rcb}(x, y)$ (Definition 3.4).
6:      **Selection**: $S' \leftarrow S' \cup \{z\}$ if $f(z) \geq \frac{f(x)+f(y)}{2}$.
7: **end while**
8: **Output:** $S'$

---

Next we turn to designing the strategy for adding directed mutation under a given $f$ as guidance and propose Module 2. An ideal mutation will diversify the population while preserving its fitness level as much as possible. So we add directed mutation to sites where the single site fitness over the population is less than of a uniformly distributed sequence. Formally, we only add mutation to site $i$ if $\frac{1}{M} \sum_{x \in S} \theta_i \cdot x_i \leq \theta_i \cdot \bar{x}_i$, where $\bar{x}_i$ is the mean of uniformly random $x_i$.

---

**Module 2** $\mathtt{Directed\_Mutation}(f, S, \mu)$

---

1: **Inputs:** utility function $f(x) = \langle \theta, x \rangle$, a population of sequences $S$, mutation rate $\mu$
2: **Initialization:** $\mathcal{I} \leftarrow \emptyset, \mathcal{S}' \leftarrow \emptyset$
3: **for** $i \in [d]$ **do**
4:      **if** $\frac{1}{M} \sum_{x \in S} \theta_i \cdot x_i \leq \theta_i \cdot \bar{x}_i$ **then**
5:         $\mathcal{I} \leftarrow \mathcal{I} \cup \{i\}$.
6:      **end if**
7: **end for**
8: **Directed Mutation:** $x' = \mathrm{Mut}(x, \mathcal{I}, \mu)$ (Definition 3.3) and $S' \leftarrow S' \cup \{x'\}$ for all $x \in S$.
9: **Output:** $S'$

---

## 4.2 Full algorithm

Finally, we are ready to combine all modules and state the full algorithm in Algorithm 1. At each time step $t$, a posterior distribution is first computed using the data collected in history. Then we sample a $\widetilde{\theta}_t$ from the posterior and do the corresponding directed mutation and crossover selection using this sampled weight, and augment the dataset for the next iteration with the measurements of resulting new population. The procedure is repeated until the time limit $T$ is reached.

# 5 Main results

In this section, we analyze the performance of TS-DE (Algorithm 1). We will show that the crossover selection module essentially mimics an optimization iteration that strictly improves the population's

---
**Algorithm 1** Thompson Sampling-Guided Directed Evolution (TS-DE)
---
1: **Inputs:** number of rounds $T$, initial population $S_0 = \{x_{0,i}\}_{i=1}^M$ of size $M$, mutation rate $\mu, \sigma$
2: **Initialization:** dataset $D_0 \leftarrow \emptyset$, $\Phi_{-1} = 0$, $U_0 = 0$
3: **for** $t = 1$ to $T$ **do**
4:     **Posterior update**

$$V_t = \frac{1}{\sigma^2} \Phi_{t-1}^\top \Phi_{t-1} + \lambda I, \qquad \widehat{\theta}_t = \frac{1}{\sigma^2} V_t^{-1} \Phi_{t-1}^\top U_{t-1}. \tag{4.1}$$

5:     **Thompson Sampling**  $\widetilde{\theta}_t \sim \mathcal{N}(\widehat{\theta}_t, V_t^{-1})$.
6:     $S'_{t-1} = \texttt{Directed\_Mutation}(f_{\widetilde{\theta}_t}, S_{t-1}, \mu)$ (Module 2).
7:     $S_t = \texttt{Crossover\_Selection}(f_{\widetilde{\theta}_t}, S'_{t-1})$ (Module 1).
8:     **Evaluation and data collection** Evaluate the utilities of all individuals in $S_t$ and $D_t \leftarrow D_{t-1} \cup \{x_{t,i}, u(x_{t,i})\}_{i=1}^M$. Update $\Phi_t^\top \leftarrow (\Phi_{t-1}^\top, x_{t,1}, \cdots, x_{t,M})$, $U_t \leftarrow (U_{t-1}^\top, u(x_{t,1}), \cdots, u(x_{t,M}))^\top$.
9:     $t \leftarrow t + 1$.
10: **end for**
---

fitness along the designated direction. By using a Bayesian regret analysis, we show the DE modules, when combined with posterior sampling, can effectively optimize towards the best protein design while learning $\theta^\star$.

## 5.1 Crossover selection as an optimization iteration

Let $f$ by any utility function, and let $F(S) := \text{avg}_{x \in S} f(x)$ denote the population average utility. Our first result states an ascent property showing that $\texttt{Crossover\_Selection}$ strictly improves the population average.

**Theorem 5.1** (Ascent Property of Recombination-then-Selection). Let $f(x) = \langle \theta, x \rangle$ and let $S$ be a set of sequences. Let $S' = \texttt{Crossover\_Selection}(f, S)$, then it satisfies

$$\mathbb{E}[F(S')] \geq F(S) + \frac{\mathbb{E}_{x,y}[\|\theta \cdot (x - y)\|]}{2\sqrt{2}} \geq F(S) + \frac{1}{\sqrt{2d}} \sum_i |\theta_i| \text{Var}_i(S), \tag{5.1}$$

where $\text{Var}_i(S)$ denotes the variance of $x_i$ when $x$ is uniformly sampled from $S$.

**Proof sketch.** See Figure 5.1 for illustration. Given $x$ and $y$, $z = \texttt{Rcb}(x, y)$ can be represented by $z = \frac{x+y}{2} + \frac{x-y}{2} \cdot e$, where the $\cdot$ denotes the entrywise multiplication between two vectors and $e = (e_i, \cdots, e_d)$ with $e_i$'s being independent Rademacher variables. Then $f(z)$ equals $\frac{f(x)+f(y)}{2} + \frac{1}{2} \sum_{i=1}^d \theta_i (x_i - y_i) e_i$. After the selection step, the expected amount by which $f(z)$ exceeds its parents' average is at least $\frac{1}{2} \mathbb{E}\left[ \left| \sum_{i=1}^d \theta_i (x_i - y_i) e_i \right| \right]$, which has a tight lower bound of $\frac{1}{2\sqrt{2}} \|\theta \cdot (x - y)\|$ according to Haagerup [20]. The full proof is given in Appendix C.1. ∎

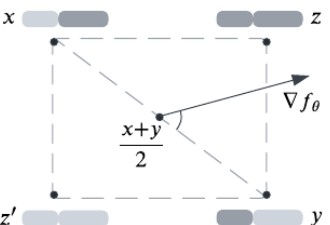

Figure 5.1: Ascent property of crossover recombination

**Remark on diversity.** Analysis above reveals an intriguing observation: the optimization progress of $\texttt{Crossover\_Selection}$ scales linearly with $\sum_i \theta_i \text{Var}_i(S)$, i.e., sum of per-motif variances across population $S$. It measures a level of "diversity" of $S$ with respect to direction $\theta$. More diverse population would enjoy larger progress from crossover selection. This observation is consistent with the natural evolution theory that diversity is key to the adaptability of a population to cope with evolving environment where fitness traits are essential [52].

## 5.2 Regret bound of TS-DE

Our main result is a Bayesian regret bound for TS-DE. Recall from Definition 3.6 that $\text{BayesRGT}(T, M) = \mathbb{E}[\sum_{t=1}^{T} \sum_{i=1}^{M}(f_{\theta^\star}(x^\star) - f_{\theta^\star}(x_{t,i}))]$.

**Theorem 5.2.** Under Assumption 3.2 and 3.5, when the population size is sufficient s.t. $M = \Omega\left(\frac{\log(dT)}{\mu^2}\right)$, Alg. 1 admits its Bayesian regret s.t.

$$\text{BayesRGT}(T, M) = \widetilde{O}\left(\frac{d}{\mu\sqrt{\lambda}} \cdot d\sqrt{MT}\right). \tag{5.2}$$

If we let $\lambda = 1, \mu = 1/2, \sigma^2 = 1$, the Bayesian regret simplifies to $\widetilde{O}(d^2\sqrt{MT})$.

**Remark on regret bound.** Regret bound of Theorem 5.2 is optimal in $M, T$. For comparison, the Bayesian regret of Gaussian linear model is $\widetilde{O}(d\sqrt{T})$ [31], also in contextual linear bandit with batch update, the optimal regret is $\widetilde{O}(d\sqrt{MT})$ [22]. Our TS-DS regret has two extra factors of $\sqrt{d}$. One $\sqrt{d}$ is due to that the $l_2$ norm of our feature vectors are $\sqrt{d}$, while linear bandit theory often assumes feature to have norm 1. Another factor of $\sqrt{d}$ is due to the evolutionary nature of DE, i.e., TS-DE is not allowed to any possible action but have to select those from the evolving population.

## 5.3 Proof sketch

**Main challenge.** Classic bandit method/analysis does not apply to our setting, because each round of DE is limited to actions that are reachable by mutation and recombination based on the current population. It means that we cannot simply explore the optimistic actions that maximize each function estimate $f_{\widetilde{\theta}_t}$. This leads to an optimization gap that complicates the regret proof.

Denote by $x^\star$ and $x_t^\star$ the maximums of $f_{\theta^\star}$ and $f_{\widetilde{\theta}_t}$. Denote by $F_t^\star := f_{\widetilde{\theta}_t}(x_t^\star)$ the maximum value of $f_{\widetilde{\theta}_t}$ and denote by $F_t(S)$ the average value of $f_{\widetilde{\theta}_t}$ over set $S$.

**Step 1: Regret decomposition.** With expectation taken over all stochasticity, posterior sampling guarantees $\text{BayesRGT}(T, M) = \sum_{t=1}^{T} \sum_{i=1}^{M} \mathbb{E}\left[f_{\widetilde{\theta}_t}(x_t^\star) - f_{\theta^\star}(x_{t,i})\right]$ since conditioned on data $D_{t-1}$, $f_{\theta^\star}(x^\star)$ and $f_{\widetilde{\theta}_t}(x_t^\star)$ are identically distributed. Then by breaking $f_{\widetilde{\theta}_t}(x_t^\star) - f_{\theta^\star}(x_{t,i})$ down to the sum of $f_{\widetilde{\theta}_t}(x_t^\star) - f_{\widetilde{\theta}_t}(x_{t,i})$ and $f_{\widetilde{\theta}_t}(x_{t,i}) - f_{\theta^\star}(x_{t,i})$, we decompose the total regret into

$$\text{BayesRGT}(T, M) = M \cdot \underbrace{\mathbb{E}\left[\sum_{t=1}^{T}(F_t^\star - F_t(S_t))\right]}_{H_1} + \underbrace{\mathbb{E}\left[\sum_{t=1}^{T}\sum_{i=1}^{M}\langle\widetilde{\theta}_t - \theta^\star, x_{t,i}\rangle\right]}_{H_2}. \tag{5.3}$$

**Step 2: Bounding $H_1$ using linear convergence.** $H_1$ is the accumulated optimization error under a time-varying objective $f_{\widetilde{\theta}_t}$. After calling $S'_{t-1} = \texttt{Directed\_Mutation}(f_{\widetilde{\theta}_t}, S_{t-1}, \mu)$ and $S_t = \texttt{Crossover\_Selection}(f_{\widetilde{\theta}_t}, S'_{t-1})$ at step $t$, the ascent property (5.1) together with property of the mutation module yields a linear convergence towards $F_t^\star$, i.e., $\mathbb{E}\left[F_t^\star - F_t(S_t) \mid S_{t-1}, \widetilde{\theta}_t\right] \leq \gamma(F_t^\star - F_t(S_{t-1}))$ with a modulus of contraction $\gamma \in (0, 1)$ s.t. $\frac{1}{1-\gamma} = O\left(\frac{\sqrt{d}}{\mu}\right)$. It follows that

$$F_t^\star - F_t(S_t) \leq \gamma\left[F_{t-1}^\star - F_{t-1}(S_{t-1})\right] + \text{error terms} + e_t,$$

where $e_t$ is a martingale difference. Applying the above recursively to $H_1$, we get $H_1 \leq$

$$\underbrace{\frac{1}{1-\gamma} \cdot \mathbb{E}\left[F_1^\star - F_1(S_0)\right]}_{O(\frac{1}{1-\gamma})} + \underbrace{\mathbb{E}\left[\sum_{k=2}^{T}\gamma^{T-k+1}F_k^\star - \gamma^{T-1}F_1^\star\right]}_{O(\frac{1}{1-\gamma})} + \underbrace{\mathbb{E}\left[\sum_{t=1}^{T}\sum_{k=1}^{t-1}\gamma^{t-k}\left(F_k(S_k) - F_{k+1}(S_k)\right)\right]}_{\kappa},$$

which is dominated by term $\kappa$ and $M \cdot \kappa \leq \frac{1}{1-\gamma} \cdot \sum_{t=1}^{T-1}\sum_{i=1}^{M}\left|\langle\widetilde{\theta}_t - \widetilde{\theta}_{t+1}, x_{t,i}\rangle\right| = O\left(\frac{1}{1-\gamma}H_2\right)$.

**Step 3: Bounding $H_2$.** $H_2$ is the accumulated prediction error of $\widetilde{\theta}_t$, which is a classic term to bound in bandit literature and is of $\widetilde{O}\left(d^{1.5}\sqrt{MT}\right)$ by using a batched self-normalization bound. ∎

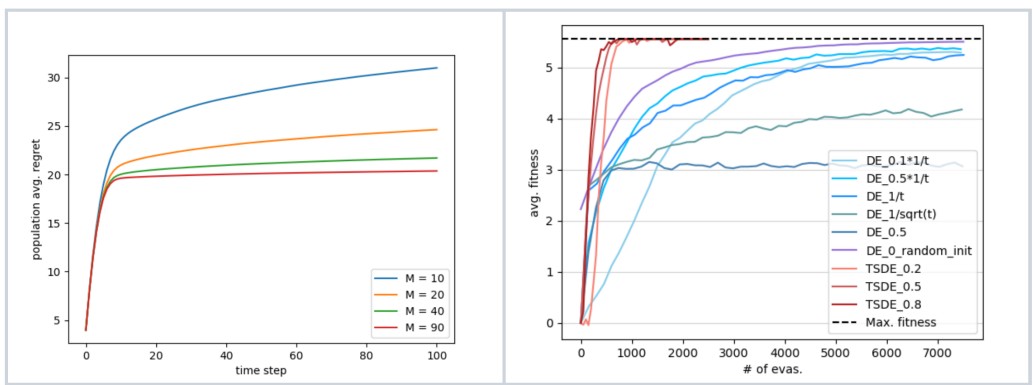

Figure 6.1: **Regret and fitness curves of TS-DE during evolution.** Left: Population-averaged regret with varying population sizes $M$. Each curve is averaged over 100 trials. Right: Fitness curves of TS-DE with varying values of $\mu$, compared with basic DE with varying mutation rates. (The purple curve plots basic DE without mutation, we modified the initial population to be uniformly distributed in this case to make it non-trivial.)

# 6 Experiments

## 6.1 Simulation

We test the TS-DE by simulating the evolution of a population of sequences in $\{0, 1\}^d$. We set the initial population to be all zeros, and set $\lambda = 1$, $\sigma = 1$.

**Regret and convergence results.** Figure 6.1 shows the regret curves and learning curves of TS-DE, with comparison to basic DE. In the left panel of Figure 6.1, we plot the population-averaged Bayesian regret of TS-DE with various values of $M$, where $d = 10$, $T = 100$ and $\mu = 0.8$. These results confirm our sublinear regret bounds. In the right panel of Figure 6.1, we tested TS-DE using various mutation rates, and compared them with a basic DE approach [5]. The comparison shows that TS-DE converges significantly faster, while the convergence of DE is much slower and very sensitive to mutation scheduling.

**Visualizing the evolution of a population.** We visualize the evolution trajectory of population $S_t$ in one run of TS-DE, with $d = 40$, $M = 20$ and $\mu = 0.1$. In the left panel of Fig 6.2, we visualize the evolving high-dimensional population $S_t$ by mapping them to 2D (via PCA and KDE density contour plot). In the right panel of Fig 6.2, we plot the fitness distribution of each $S_t$. These plots illustrate how TS-DE balances the exploration-exploitation trade-off: It guides $S_t$ to "diversify" initially and then quickly approach and concentrate around a maximal solution.

## 6.2 Real-world experiment validation

Having demonstrated our approach with simulations, we use real-world experiments to showcase the validity and generalizability of our method. The TS-DE method is adapted to work with real-world motif features (continuous-valued instead of binary), linear model and multiple rounds of wet-lab experiments for optimizing a CRISPR design sequence. Our approach together with high-throughput experiment identified a high-performing sequence with 30+ fold improvement in efficiency. Notably, the optimized CRISPR designs generated by our DE approach was experimentally validated in [26] and demonstrated the real-world utility of our method. This technology is used for ex-vivo high-throughput single-cell barcoding with applications in genomics and drug discovery.

We postpone more details about this real-world validation to Appendix B.1 and Figure B.1.

---

[5]The basic DE approach does not employ any function estimate. It does random mutation with a predefined mutation rate and random crossover recombination. It evaluates every candidate sequence and uses the noisy feedback in replace of $f_{\tilde{\theta}}$ for selection.

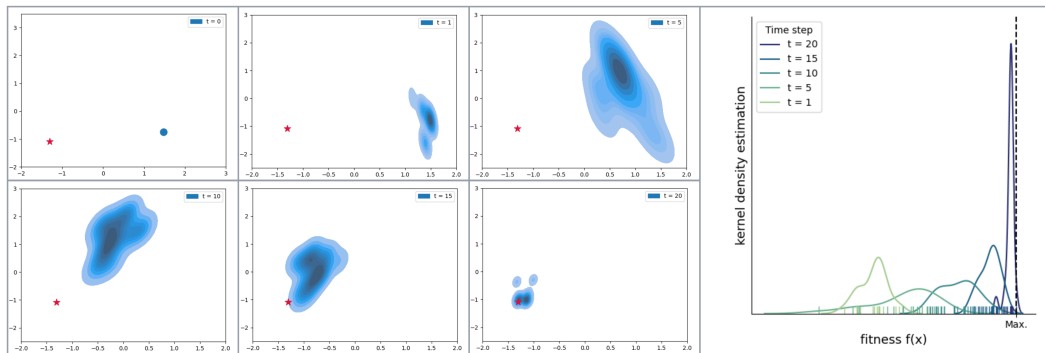

Figure 6.2: **Evolving population of TS-DE and fitness levels.** Left panels: Visualization of population evolution projected in 2D shown, taken at 6 snapshots. Right panel: The population's fitness distribution shifts towards optimal during evolution. ⋆ denotes the optimal solution.

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
