# A Proof of Theorem 5.2

## A.1 Notations

We address the following notations that frequently occur throughout the proof section. Denote by $f$ an arbitrary linear fitness function $f(x) := \langle \theta, x \rangle$ parameterized by some $\theta \in \mathbb{R}^d$ and denote by $F^\star$ its maximum. Define $F(S) := \mathrm{avg}_{x \in S} f(x)$, the average fitness under $f$ of population $S$. While $f$ represents arbitrary fitness function, $\{f_{\widetilde{\theta}_t}(x) := \langle \widetilde{\theta}_t, x \rangle\}_{t \in [T]}$ are the linear function parameterized by $\{\widetilde{\theta}_t\}_{t \in [T]}$ obtained by posterior sampling in each iteration of Alg.1. Corresponding to each $f_{\widetilde{\theta}_t}$, $F_t^\star := f_{\widetilde{\theta}_t}(x_t^\star)$ is its maximum value and $x_t^\star$ is its one maximum point. Denote by $F_t(S)$ the average $f_{\widetilde{\theta}_t}$ value over $S$. For a clear display, denote by $L$, an upper bound for the $l_2$ norm of any $x_{t,i}$ evaluated, i.e. $\|x_{t,i}\| \leq L$ and in our setting, take $L = \sqrt{d}$. Without clarification $\|\cdot\|$ denotes the $l_2$ norm by default and $\|\cdot\|_A$ denotes the norm normalized by matrix $A$.

## A.2 Routine of Alg.1 and filtrations

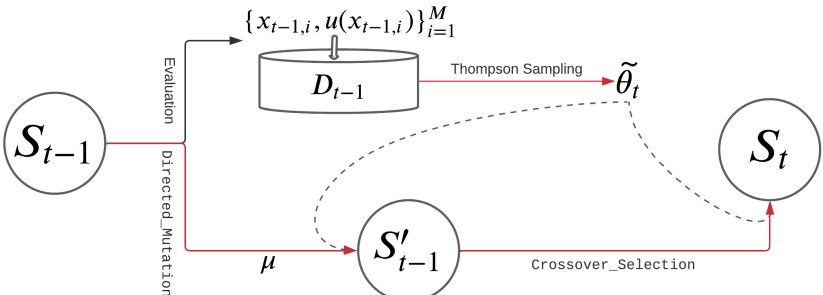

Figure A.1: **Routine of Alg.1.** Red lines represent stochastic steps. Dash lines indicate $\widetilde{\theta}_t$ is involved in those steps.

In Alg.1, there are three steps introducing stochasticity. Two of which are calling Module 2 as $S'_{t-1} = \texttt{Directed\_Mutation}(f_{\widetilde{\theta}_t}, S_{t-1}, \mu)$ and calling as $S_t = \texttt{Crossover\_Selection}(f_{\widetilde{\theta}_t}, S'_{t-1})$. Another one is Thompson sampling step s.t. $\widetilde{\theta}_t$ is sampled from the posterior of $\theta^\star$ given data $D_{t-1}$. Fig. A.1 illustrates how these three steps are built into the algorithm routine.

There are two other sources of stochasticity inherited from the problem setting: the prior of $\theta^\star$ (Assumption 3.2) and the noisy feedback $\{u(x_{t,i})\}_{i=1}^M$ (Assumption 3.5), which are revealed in the evaluation step. Including all stochasticity, the trajectory of Alg.1 is

$$\theta^\star, \widetilde{\theta}_1, S'_0, S_1, \{u(x_{1,i})\}_{i=1}^M, \cdots, \widetilde{\theta}_{t+1}, S'_t, S_{t+1}, \{u(x_{t+1,i})\}_{i=1}^M, \cdots, \widetilde{\theta}_T, S'_{T-1}, S_T, \{u(x_{T,i})\}_{i=1}^M. \tag{A.1}$$

At the convenience of analysis, we introduce multiple lines of the history up to time step $t$ by carefully partitioning the trajectory (A.1), using $\sigma(\cdot)$ to represent the minimal sigma algebra expanded by $\cdot$.

**Definition A.1.** Define a filtration $\{\mathcal{H}_t^M\}_{t=0}^{T-1}$ with $\mathcal{H}_t^M$ be the information accumulated after $t$ rounds of Alg.1 but before the Directed Mutation step in round $t+1$.

$$\mathcal{H}_0^M := \sigma\left(\theta^\star, \widetilde{\theta}_1\right),$$
$$\mathcal{H}_t^M := \left(\mathcal{H}_{t-1}^M, \sigma\left(S'_{t-1}, S_t, \{u(x_{t,i})\}_{i=1}^M, \widetilde{\theta}_{t+1}\right)\right), \quad t \in [T-1].$$

**Definition A.2.** Define a filtration $\{\mathcal{H}_t^R\}_{t=0}^{T-1}$ with $\mathcal{H}_t^R$ be the information accumulated after $t$ rounds of Alg.1 but before the Recombination and Selection step in round $t+1$.

$$\mathcal{H}_0^R := \sigma\left(\theta^\star, \widetilde{\theta}_1, S'_0\right),$$
$$\mathcal{H}_t^R := \left(\mathcal{H}_{t-1}^R, \sigma\left(S_t, \{u(x_{t,i})\}_{i=1}^M, \widetilde{\theta}_{t+1}, S'_t\right)\right), \quad t \in [T-1].$$

## A.3  Property of `Directed_Mutation` (Module 2)

Given a fitness function $f(x) := \langle \theta, x \rangle$, a useful observation is that for the dimension where $\theta_i \geq 0$, feature value $1$ is more favorable than $0$ in terms of a higher fitness. So in a population $S$, for each dimension $i$, the ratio of individuals who are with the favored feature is a key quantity, and we define it formally as follows.

**Definition A.3** (Ratio of the Favored Feature)**.** Under a fitness function $f(x) := \langle \theta, x \rangle$, for a population $S$, define

$$p_i^\theta(S) = \begin{cases} \frac{\#\{x \in S : x_i = 0\}}{|S|} & \theta_i < 0 \\ \frac{\#\{x \in S : x_i = 1\}}{|S|} & \theta_i \geq 0, \end{cases} \quad \forall i \in [d], \tag{A.2}$$

and we are allowed to omit the superscript $\theta$ of $p_i^\theta(S)$ when $\theta$ is clear from the context.

We show the following property of `Directed_Mutation`.

**Lemma A.4.** Suppose $S' = $ `Directed_Mutation`$(f, S, \mu)$, then the population-averaged fitness of $S'$ will not decrease compared to that of $S$ in expectation, that is,

$$\mathbb{E}\left[F(S')\right] \geq F(S). \tag{A.3}$$

And for $\forall \delta \in (0, 1)$, if $|S| = \Omega\left(\frac{\log(\frac{d}{\delta})}{\mu^2}\right)$, then with probability $1 - \delta$,

$$p_i\left(S'\right) \geq \frac{\mu}{4}, \quad \forall i \in [d]. \tag{A.4}$$

*Proof.* See Appendix D.1. □

### A.3.1   High probability events on $\min_i p_i^{\widetilde{\theta}_{t+1}}(S'_t)$

The Directed Mutation step of Alg.1 ensures $S'_t$ is always sufficient with the feature favored by current $f_{\widetilde{\theta}_{t+1}}$ in each dimension $i$ throughout $T$ rounds, i.e. $\min_i p_i^{\widetilde{\theta}_{t+1}}(S'_t)$ is lower bounded for $\forall t + 1 \in [T]$, recall Definition A.3 for the definition of $p_i^\theta(S)$.

We introduce the following line of events where this sufficiency of $S'_t$ holds and show the intersection of them happens with high probability when the population size $M$ is sufficiently large.

**Definition A.5.** Define $E_t^{\mathrm{DM}}$ to be the event where $\min_i p_i^{\widetilde{\theta}_{s+1}}(S'_s)$ is lower bounded by $\frac{\mu}{4}$ for $\forall s \leq t$, that is

$$E_t^{\mathrm{DM}} := \{\forall s \leq t, \quad \min_i p_i^{\widetilde{\theta}_{s+1}}(S'_s) \geq \frac{\mu}{4}\}. \tag{A.5}$$

Also define $E_{\mathrm{DM}}$ as the intersection of $\{E_t^{\mathrm{DM}}\}_{t=0}^{T-1}$.

**Lemma A.6.** For $\forall \delta \in (0, 1)$, if the population size is sufficiently large s.t. $M = O\left(\frac{\log(\frac{dT}{\delta})}{\mu^2}\right)$, then

$$\mathbb{P}\left(E_{\mathrm{DM}}\right) \geq 1 - \delta. \tag{A.6}$$

Since event $E_{\mathrm{DM}}$ is independent from the realization of $\theta^\star$, thus it still holds with high probability when conditioned on $\theta^\star$. Denote by $E_{\mathrm{DM}}^{\theta^\star}$, the event $E_{\mathrm{DM}}$ conditioned on $\theta^\star$, then

$$\mathbb{P}\left(E_{\mathrm{DM}}^{\theta^\star}\right) \geq 1 - \delta.$$

*Proof.* See Appendix D.2. □

## A.4  Linear convergence of `Crossover_Selection` (Module 1)

Continuing from Theorem 5.1, when $S$ is sufficient with the feature favored by $f$ in every dimension, i.e. $\min_i p_i(S)$ is lower bounded, then in expectation, $F(S')$ converges linearly to $F^\star$ with a nontrivial convergence rate.

**Lemma A.7** (Linear Convergence). Suppose $S' = \texttt{Crossover\_Selection}(f, S)$, then

$$\mathbb{E}[F(S')] \geq F(S) + \eta (F^\star - F(S)), \tag{A.7}$$

with factor $\eta = \frac{\min_i p_i(S)}{\sqrt{2d}}$.

*Proof.* See Appendix D.3. $\qquad\square$

### A.5 Thompson Sampling

According to Assumption 3.5, in the dataset $D_t = \left\{ \{x_{s,i}, u(x_{s,i})\}_{i=1}^M, s = [t] \right\}$

$$u(x_{s,i}) = f_{\theta^\star}(x_{s,i}) + \xi_{s,i}, \tag{A.8}$$

with $\xi_{s,i}$ i.i.d. sampled from $\mathcal{N}(0, \sigma^2)$ and independent from all other stochasticity.

Therefore, by Bayes' Rule, the posterior of $\theta^\star$ give $D_{t-1}$ is also Gaussian distributed, for $\forall t \in [T]$:

$$\widetilde{\theta}_t \sim \mathcal{N}(\widehat{\theta}_t, V_t^{-1}), \tag{A.9}$$

where

$$V_t = \frac{1}{\sigma^2} \Phi_{t-1}^\top \Phi_{t-1} + \lambda I, \tag{A.10}$$

$$\widehat{\theta}_t = \frac{1}{\sigma^2} V_t^{-1} \Phi_{t-1}^\top U_{t-1}, \tag{A.11}$$

recall from Alg. 1 for the updating rules of $\Phi_t$ and $U_t$.

Given the posterior distribution (A.9), we are able to show $\widetilde{\theta}_t$ concentrates to $\theta^\star$ in term of the normalized distance between them.

### A.5.1 High probability events on $\left\| \widetilde{\theta}_t - \theta^\star \right\|_{V_t}$

We introduce two useful lines of high probability events similar to those in Abeille and Lazaric [2], except here these events are defined conditioned on any realization of $\theta^\star$ sampled from its prior. We rephrased the definition to match our notations.

**Definition A.8.** Conditioned on $\theta^\star$, for any given probability tolerance $\delta \in (0, 1)$, each time step $t \in [T]$ and a line of ($\theta^\star$ dependent) radiuses $\{\beta_t^{\theta^\star}(\delta)\}_{t=1}^T$, we define $\widehat{E}_t^{\theta^\star}$ as the event where $\widehat{\theta}_s$ concentrates around $\theta^\star$ for all previous steps $s \leq t$, i.e.

$$\widehat{E}_t^{\theta^\star}(\delta) = \left\{ \forall s \leq t, \left\| \widehat{\theta}_s - \theta^\star \right\|_{V_s} \leq \beta_s^{\theta^\star}(\delta) \,\middle|\, \theta^\star \right\}. \tag{A.12}$$

with a line of ($\theta^\star$ independent) radiuses $\{\alpha_t(\delta)\}_{t=1}^T$, we also define $\widetilde{E}_t^{\theta^\star}$ as the event where the sampled parameter $\widetilde{\theta}_s$ concentrates around $\widehat{\theta}_s$ for all steps $s \leq t$, i.e.

$$\widetilde{E}_t^{\theta^\star}(\delta) = \left\{ \forall s \leq t, \left\| \widetilde{\theta}_s - \widehat{\theta}_s \right\|_{V_s} \leq \alpha_s(\delta) \,\middle|\, \theta^\star \right\}. \tag{A.13}$$

Then under the same $\delta$ and $\theta^\star$, which are omitted here, we have $\widehat{E} := \widehat{E}_T \subset \cdots \subset \widehat{E}_1$, $\widetilde{E} := \widetilde{E}_T \subset \cdots \subset \widetilde{E}_1$ and define $E^{\theta^\star}(\delta) := \widehat{E}^{\theta^\star}(\delta) \cap \widetilde{E}^{\theta^\star}(\delta)$.

With appropriate choices of $\{\beta_t\}$ and $\{\alpha_t\}$, event $E^{\theta^\star}(\delta)$ defined above happens with high probability as stated in the following lemma.

**Lemma A.9.** Under Assumption 3.2 and 3.5, conditioned on any realization of $\theta^\star$ drawn from its prior, for $\forall \delta \in (0, 1)$ and any series of feature vectors $\left( \{x_{1,i}\}_{i=1}^M, \cdots, \{x_{T,i}\}_{i=1}^M \right)$ where each $\|x_{t,i}\| \leq L$, $\mathbb{P}\left( E^{\theta^\star}\left(\frac{\delta}{2}\right) \right) \geq 1 - \delta$ with $\beta_t^{\theta^\star}(\delta)$ and $\alpha_t(\delta)$ specified as

$$\beta_t^{\theta^\star}(\delta) = \sqrt{2 \log\left(\frac{1}{\delta}\right) + d \log\left(\frac{\sigma^2 \lambda d + t M L^2}{\sigma^2 \lambda d}\right)} + \sqrt{\lambda} \|\theta^\star\|, \quad \forall t \in [T]. \tag{A.14}$$

$$\alpha_t(\delta) = 2\sqrt{d \log\left(\frac{T}{\delta}\right)} + \sqrt{d}, \quad \forall t \in [T]. \tag{A.15}$$

*Proof.* See Appendix D.4. $\qquad\square$

## A.6 Prediction error under batch update

Before regret decomposition, one more preparation to have is a modified concentration on the accumulated prediction error of $\widetilde{\theta}_t$ catering for the batch-data update routine in Alg.1. In the following lemma, we summarize a more general version of this concentration result.

**Lemma A.10.** Suppose at any timestep $a_t$ lies in a confidence ellipsoid around $b_t$ in the sense that

$$\|a_t - b_t\|_{V_t} \leq \eta_t(\delta), \quad \forall t \in [T],$$

and $\|x_{t,i}\| \leq L, \forall t \in [T], i \in [M]$, then it holds that,

$$\sum_{t=1}^{T} \sum_{i=1}^{M} |\langle a_t - b_t, x_{t,i} \rangle|$$

$$\leq \eta_T(\delta) \sqrt{\frac{2L^2 + 2\lambda}{\lambda}} \cdot \sqrt{dMT \log\left(\frac{\sigma^2 d\lambda + MTL^2}{\sigma^2 d\lambda}\right)} + \eta_T(\delta) \frac{2L}{\sqrt{\lambda}} \cdot dM \log\left(\frac{\sigma^2 d\lambda + MTL^2}{\sigma^2 d\lambda}\right).$$

$$\text{(A.16)}$$

And let us give an alias RGT $(\eta_T(\delta))$ to the RHS of (A.16).

*Proof.* See Appendix D.6. □

## A.7 Regret decomposition

Recall the notation that $x^\star$ is a maximum point of $f_{\theta^\star}$.

By the scheme of posterior sampling, $f_{\theta^\star}(x^\star)$ and $f_{\widetilde{\theta}_t}(x_t^\star)$ are identically distributed conditioned on $D_{t-1}$, which leads to

$$\mathbb{E}\left[f_{\theta^\star}(x^\star) - f_{\widetilde{\theta}_t}(x_t^\star)\middle| D_{t-1}\right] = 0. \tag{A.17}$$

With expectation taken over all stochasticity, the per-round Bayesian regret is

$$\mathbb{E}\left[\sum_{i=1}^{M}(f_{\theta^\star}(x^\star) - f_{\theta^\star}(x_{t,i}))\right] = \sum_{i=1}^{M}\mathbb{E}\left[f_{\theta^\star}(x^\star) - f_{\widetilde{\theta}_t}(x_t^\star)\right] + \sum_{i=1}^{M}\mathbb{E}\left[f_{\widetilde{\theta}_t}(x_t^\star) - f_{\theta^\star}(x_{t,i})\right]$$

$$= \sum_{i=1}^{M}\mathbb{E}\left[\mathbb{E}\left[f_{\theta^\star}(x^\star) - f_{\widetilde{\theta}_t}(x_t^\star)\middle| D_{t-1}\right]\right] + \sum_{i=1}^{M}\mathbb{E}\left[f_{\widetilde{\theta}_t}(x_t^\star) - f_{\theta^\star}(x_{t,i})\right]$$

$$\overset{(A.17)}{=} \sum_{i=1}^{M}\mathbb{E}\left[f_{\widetilde{\theta}_t}(x_t^\star) - f_{\theta^\star}(x_{t,i})\right]$$

$$= \sum_{i=1}^{M}\mathbb{E}\left[f_{\widetilde{\theta}_t}(x_t^\star) - f_{\widetilde{\theta}_t}(x_{t,i}) + f_{\widetilde{\theta}_t}(x_{t,i}) - f_{\theta^\star}(x_{t,i})\right]$$

$$= \mathbb{E}\left[M\left(F_t^\star - F_t(S_t)\right)\right] + \mathbb{E}\left[\sum_{i=1}^{M}\langle\widetilde{\theta}_t - \theta^\star, x_{t,i}\rangle\right].$$

Then the total Bayesian regret over $T$ rounds sums up to be

$$\text{BayesRGT}(T, M) = \mathbb{E}\left[M\sum_{t=1}^{T}F_t^\star - F_t(S_t)\right] + \mathbb{E}\left[\sum_{t=1}^{T}\sum_{i=1}^{M}\langle\widetilde{\theta}_t - \theta^\star, x_{t,i}\rangle\right]$$

$$= \mathbb{E}_{\theta^\star\sim\pi}\left[\mathbb{E}_{\theta^\star}\left[M\sum_{t=1}^{T}F_t^\star - F_t(S_t)\right]\right] + \mathbb{E}_{\theta^\star\sim\pi}\left[\mathbb{E}_{\theta^\star}\left[\sum_{t=1}^{T}\sum_{i=1}^{M}\langle\widetilde{\theta}_t - \theta^\star, x_{t,i}\rangle\right]\right],$$

$$\text{(A.18)}$$

where $\mathbb{E}_{\theta^\star}[\cdot]$ denotes the conditional expectation on a given $\theta^\star$: $\mathbb{E}[\cdot|\theta^\star]$.

Note that under any realization of $\theta^\star$, the regret of each individual at any time step should be no more than the range of $f_{\theta^\star}$ on domain $\mathcal{X}$. For any $\theta \in R^d$ parameterizing the fitness $f_\theta$ as $f_\theta(x) = \langle \theta, x \rangle$, denote by $B_f^\theta$ an upper bound for the range of $f_\theta$, i.e.

$$B_f^\theta := 2L\|\theta\| \geq \max_x f_\theta(x) - \min_x f_\theta(x). \tag{A.19}$$

For the regret of each individual in each step, it holds that

$$f_{\theta^\star}(x^\star) - f_{\theta^\star}(x_{t,i}) \leq 2\|\theta^\star\|L = B_f^{\theta^\star}. \tag{A.20}$$

Therefore, when bounding the total regret decomposed as (A.18), it is reasonable to truncate terms in the RHS of (A.18) with $B_f^{\theta^\star}$ to derive a tighter bound.

$$\text{BayesRGT}(T, M) \leq \mathbb{E}_{\theta^\star \sim \pi} \left[ M\mathbb{E}_{\theta^\star} \left[ \sum_{t=1}^T \min \left\{ F_t^\star - F_t(S_t), B_f^{\theta^\star} \right\} \right] \right] \tag{A.21}$$

$$+ \mathbb{E}_{\theta^\star \sim \pi} \left[ \mathbb{E}_{\theta^\star} \left[ \sum_{t=1}^T \sum_{i=1}^M \min \left\{ \left| \langle \widetilde{\theta}_t - \theta^\star, x_{t,i} \rangle \right|, B_f^{\theta^\star} \right\} \right] \right] \tag{A.22}$$

## A.8 Bounding the first half (A.21)

### A.8.1 After calling $S'_{t-1} = \texttt{Directed\_Mutation}(f_{\widetilde{\theta}_t}, S_{t-1}, \mu)$

As shown in Lemma A.4, the population average of $S'_t$ under $f_{\widetilde{\theta}_{t+1}}$ in not decreasing from that of $S_t$, that is

$$\mathbb{E}\left[ F_{t+1}(S'_t) | \mathcal{H}_t^M \right] \geq F_{t+1}(S_t). \tag{A.23}$$

The other property of $\texttt{Directed\_Mutation}$ is to ensure that w.h.p. $\min_i p_i^{\widetilde{\theta}_{t+1}}(S'_t)$ is lower bounded for $\forall t + 1 \in [T]$, which is stated in the definition of event $E_{\text{DM}}^{\theta^\star}$ (Definition A.5). So from here on, given any realization of $\theta^\star$, our further analysis is conditioned on $E^{\theta^\star} := E^{\theta^\star}\left(\frac{\delta}{2}\right) \cap E_{\text{DM}}^{\theta^\star}$.

**Corollary A.11.** Given any realization of $\theta^\star$, if $M = \Omega\left(\frac{\log(\frac{dT}{\delta})}{\mu_M^2}\right)$, then $\mathbb{P}\left(E^{\theta^\star}\right) \geq 1 - 2\delta$ for $\forall \delta \in (0, 1)$. Conditioned on $E^{\theta^\star} := E^{\theta^\star}\left(\frac{\delta}{2}\right) \cap E_{\text{DM}}^{\theta^\star}$, it is guaranteed that

$$\min_i p_i^{\widetilde{\theta}_{t+1}}(S'_t) \geq C_{S'} := \frac{\mu}{4}, \quad \forall t + 1 \in [T], \tag{A.24}$$

$$\|\widetilde{\theta}_t - \theta^\star\|_{V_t} \leq \beta_t^{\theta^\star}\left(\frac{\delta}{2}\right) + \alpha_t\left(\frac{\delta}{2}\right), \quad \forall t \in [T]. \tag{A.25}$$

where recall the definition of $\beta_t^{\theta^\star}$ and $\alpha_t$ from (A.14) and (A.15).

*Proof.* The proof is directly derived by combining Lemma A.6 and Lemma A.9. $\qquad\square$

### A.8.2 After calling $S_t = \texttt{Crossover\_Selection}(f_{\widetilde{\theta}_t}, S'_{t-1})$

Conditioned on $\theta^\star$, we are about to decompose $\sum_{t=1}^T (F_t^\star - F_t(S_t))$ by leveraging the linear convergence property shown in Lemma A.7. Conditionally on $E^{\theta^\star}$, applying Lemma A.7 to each call of $\texttt{Crossover\_Selection}(f_{\widetilde{\theta}_t}, S'_{t-1})$ guarantees for $\forall t + 1 \in [T]$,

$$\mathbb{E}_{E^{\theta^\star}}\left[ F_{t+1}(S_{t+1}) | \mathcal{H}_t^R \right] \geq \mathbb{E}_{E^{\theta^\star}}\left[ F_{t+1}(S'_t) + \frac{\min_i p_i^{\widetilde{\theta}_{t+1}}(S'_t)}{\sqrt{2d}} \left( F_{t+1}^\star - F_{t+1}(S'_t) \right) \Big| \mathcal{H}_t^R \right]$$

$$\overset{(A.24)}{\geq} F_{t+1}(S'_t) + \frac{C_{S'}}{\sqrt{2d}} \left( F_{t+1}^\star - F_{t+1}(S'_t) \right), \tag{A.26}$$

where $\mathbb{E}_{E^{\theta^\star}}[\cdot]$ is the conditional expectation on event $E^{\theta^\star}$.

Recall (A.23) that
$$\mathbb{E}\left[F_{t+1}(S_t')\mid \mathcal{H}_t^M\right] \geq F_{t+1}(S_t). \tag{A.23 revisited}$$

Conditioned on $E^{\theta^\star}$, it still holds that
$$\mathbb{E}_{E^{\theta^\star}}\left[F_{t+1}(S_t')\mid \mathcal{H}_t^M\right] \geq F_{t+1}(S_t), \tag{A.27}$$

since in $E^{\theta^\star}$, $E^{\theta^\star}\left(\frac{\delta}{2}\right)$ holds independent from the Directed Mutation step $S_t' = \mathrm{DM}(f_{\widetilde{\theta}_{t+1}}, S_t, \mu_M)$, and conditioned on $E^{\theta^\star}_{\mathrm{DM}}$, $F_{t+1}(S_t')\mid \mathcal{H}_t^M$ tends to be greater then it was unconditionally.

Along with $\mathcal{H}_t^M \subset \mathcal{H}_t^R$, we have
$$\mathbb{E}_{E^{\theta^\star}}\left[F_{t+1}(S_{t+1})\mid \mathcal{H}_t^M\right] \geq \mathbb{E}_{E^{\theta^\star}}\left[\mathbb{E}_{E^{\theta^\star}}\left[F_{t+1}(S_{t+1})\mid \mathcal{H}_t^R\right]\mid \mathcal{H}_t^M\right] \tag{A.28}$$
$$\overset{(A.26)}{\geq} \mathbb{E}_{E^{\theta^\star}}\left[F_{t+1}(S_t') + \frac{C_{S'}}{\sqrt{2d}}\left(F_{t+1}^\star - F_{t+1}(S_t')\right)\bigg|\mathcal{H}_t^M\right] \tag{A.29}$$
$$\overset{(A.27)}{\geq} F_{t+1}(S_t) + \frac{C_{S'}}{\sqrt{2d}}\left(F_{t+1}^\star - F_{t+1}(S_t)\right). \tag{A.30}$$

By introducing the convergence rate $\gamma := 1 - \frac{C_{S'}}{\sqrt{2d}}$ s.t. $\frac{1}{1-\gamma} = O\left(\frac{\sqrt{d}}{\mu}\right)$ and an residual term
$$e_{t+1} := \mathbb{E}_{E^{\theta^\star}}\left[F_{t+1}(S_{t+1})\mid \mathcal{H}_t^M\right] - F_{t+1}(S_{t+1}),$$

we have
$$F_{t+1}^\star - F_{t+1}(S_{t+1}) \leq \gamma(F_{t+1}^\star - F_{t+1}(S_t)) + e_{t+1}, \tag{A.31}$$
where $\{e_t\}_{t=1}^T$ is a martingale difference with
$$\mathbb{E}_{E^{\theta^\star}}[e_{t+1} \mid \mathcal{H}_t^M] = 0. \tag{A.32}$$

Thus,
$$
\begin{aligned}
F_{t+1}^\star - F_{t+1}(S_{t+1}) &\leq \gamma(F_{t+1}^\star - F_{t+1}(S_t)) + e_{t+1}\\
&= \gamma\left[F_t^\star - F_t(S_t) + F_{t+1}^\star - F_t^\star + F_t(S_t) - F_{t+1}(S_t)\right] + e_{t+1}\\
&= \gamma\left[F_t^\star - F_t(S_t)\right] + \gamma\left[F_{t+1}^\star - F_t^\star + F_t(S_t) - F_{t+1}(S_t)\right] + e_{t+1}.
\end{aligned}
$$

Therefore we have the recursion that
$$F_t^\star - F_t(S_t) \leq \gamma^t\left(F_1^\star - F_1(S_0)\right) + \sum_{k=1}^{t-1}\gamma^{t-k}\left(F_{k+1}^\star - F_k^\star\right) + \sum_{k=1}^{t-1}\gamma^{t-k}\left(F_k(S_k) - F_{k+1}(S_k)\right) + \sum_{k=1}^{t}\gamma^{t-k}e_k, \tag{A.33}$$

summing up which from $t=1$ to $T$ gives
$$\sum_{t=1}^T F_t^\star - F_t(S_t) \leq \sum_{t=1}^T\sum_{k=1}^t \gamma^{t-k}e_k \tag{A.34}$$
$$+ \sum_{t=1}^T \gamma^t\left(F_1^\star - F_1(S_0)\right) \tag{A.35}$$
$$+ \sum_{t=1}^T\sum_{k=1}^{t-1}\gamma^{t-k}\left(F_{k+1}^\star - F_k^\star\right) \tag{A.36}$$
$$+ \sum_{t=1}^T\sum_{k=1}^{t-1}\gamma^{t-k}\left(F_k(S_k) - F_{k+1}(S_k)\right). \tag{A.37}$$

As it appears in (A.21), what matters in bounding regret is the expected truncated value of $\sum_{t=1}^T F_t^\star - F_t(S_t)$, which is
$$\mathbb{E}_{\theta^\star \sim \pi}\left[M\mathbb{E}_{\theta^\star}\left[\sum_{t=1}^T \min\left\{F_t^\star - F_t(S_t), B_f^{\theta^\star}\right\}\right]\right], \tag{A.21 revisited}$$

and the decomposition of $\sum_{t=1}^{T} F_t^\star - F_t(S_t)$ into four terms as above holds conditionally on $E^{\theta^\star}$. So from here on, we progress with first upper bounding $\mathbb{E}_{\theta^\star}\left[\sum_{t=1}^{T} \min\left\{F_t^\star - F_t(S_t), B_f^{\theta^\star}\right\}\right]$ by

$$\mathbb{E}_{\theta^\star}\left[\sum_{t=1}^{T} \min\left\{F_t^\star - F_t(S_t), B_f^{\theta^\star}\right\}\right] \leq 2\delta T B_f^{\theta^\star} + (1-2\delta)\mathbb{E}_{E^{\theta^\star}}\left[\sum_{t=1}^{T} F_t^\star - F_t(S_t)\right], \quad \text{(A.38)}$$

and then upper bounding $\mathbb{E}_{E^{\theta^\star}}\left[\sum_{t=1}^{T} F_t^\star - F_t(S_t)\right]$ with

$$\mathbb{E}_{E^{\theta^\star}}\left[\sum_{t=1}^{T} F_t^\star - F_t(S_t)\right] \leq \mathbb{E}_{E^{\theta^\star}}\left[\sum_{t=1}^{T}\sum_{k=1}^{t} \gamma^{t-k} e_k\right] \tag{A.39}$$

$$+ \mathbb{E}_{E^{\theta^\star}}\left[\sum_{t=1}^{T} \gamma^t \left(F_1^\star - F_1(S_0)\right)\right] \tag{A.40}$$

$$+ \mathbb{E}_{E^{\theta^\star}}\left[\sum_{t=1}^{T}\sum_{k=1}^{t-1} \gamma^{t-k} \left(F_{k+1}^\star - F_k^\star\right)\right] \tag{A.41}$$

$$+ \mathbb{E}_{E^{\theta^\star}}\left[\sum_{t=1}^{T}\sum_{k=1}^{t-1} \gamma^{t-k} \left(F_k(S_k) - F_{k+1}(S_k)\right)\right]. \tag{A.42}$$

### A.8.3 Term (A.39)

$\{e_t\}_{t=1}^{T}$ is claimed to be a martingale difference when first being introduced, that is, recall (A.32) that

$$\mathbb{E}_{E^{\theta^\star}}[e_{t+1} \mid \mathcal{H}_t^M] = 0, \quad \forall t+1 \in [T]. \tag{A.32 revisited}$$

Thus by the property of martingale difference,

$$\mathbb{E}_{E^{\theta^\star}}[e_{t+1}] = \mathbb{E}_{E^{\theta^\star}}\left[\mathbb{E}_{E^{\theta^\star}}[e_{t+1} \mid \mathcal{H}_t^M]\right] = 0, \quad \forall t+1 \in [T].$$

Then by the linearity of $\mathbb{E}_{E^{\theta^\star}}[\cdot]$:

$$\mathbb{E}_{E^{\theta^\star}}\left[\sum_{t=1}^{T}\sum_{k=1}^{t} \gamma^{t-k} e_k\right] = \sum_{t=1}^{T}\sum_{k=1}^{t} \mathbb{E}_{E^{\theta^\star}}\left[\gamma^{t-k} e_k\right]$$

$$= \sum_{t=1}^{T}\sum_{k=1}^{t} \gamma^{t-k}\mathbb{E}_{E^{\theta^\star}}[e_k]$$

$$= 0. \tag{A.43}$$

### A.8.4 Term (A.40)

Before looking into the term (A.40), we first introduce the following lemma upper bounding the expectation of $\widetilde{\theta}_t$'s $l_2$ norm conditioned on event $E^{\theta^\star}$.

**Lemma A.12.** For $\forall t \in [T]$, $\mathbb{E}_{E^{\theta^\star}}\left[\|\widetilde{\theta}_t\|\right]$ has the following upper bound.

$$\mathbb{E}_{E^{\theta^\star}}\left[\|\widetilde{\theta}_t\|\right] \leq 2\|\theta^\star\| + 2\sqrt{\frac{d}{\lambda}}. \tag{A.44}$$

*Proof.* See Appendix D.8. ∎

What is to take expectation in (A.40) is of constant order because

$$\sum_{t=1}^{T} \gamma^{t-1} \left(F_1^\star - F_1(S_0)\right) \leq \frac{1}{1-\gamma}|F_1^\star - F_1(S_0)| \leq \frac{1}{1-\gamma} B_f^{\widetilde{\theta}_1}, \tag{A.45}$$

where $B_f^{\widetilde{\theta}_1} = 2L\|\widetilde{\theta}_1\|$.

Then by taking expectation over both sides of (A.45), we have

$$\mathbb{E}_{E^{\theta^\star}}\left[\sum_{t=1}^{T}\gamma^t\left(F_1^\star - F_1(S_0)\right)\right] \le \frac{2L}{1-\gamma}\mathbb{E}_{E^{\theta^\star}}\left[\|\widetilde{\theta}_1\|\right] \le \frac{2}{1-\gamma}\left(B_f^{\theta^\star} + 2\sqrt{\frac{d}{\lambda}}L\right). \qquad \text{(A.46)}$$

### A.8.5 Term (A.41)

Rearrange terms to sum up in (A.41) as

$$\sum_{t=1}^{T}\sum_{k=1}^{t-1}\gamma^{t-k}\left(F_{k+1}^\star - F_k^\star\right) = \sum_{k=1}^{T-1}\left(F_{k+1}^\star - F_k^\star\right)\sum_{t=k+1}^{T}\gamma^{t-k}$$

$$= \sum_{k=1}^{T-1}\frac{\gamma - \gamma^{T-k+1}}{1-\gamma}\left(F_{k+1}^\star - F_k^\star\right)$$

$$= \frac{\gamma}{1-\gamma}(F_T^\star - F_1^\star) - \sum_{k=1}^{T-1}\frac{\gamma^{T-k+1}}{1-\gamma}\left(F_{k+1}^\star - F_k^\star\right)$$

$$= \frac{\gamma}{1-\gamma}(F_T^\star - F_1^\star) - \frac{\gamma^2}{1-\gamma}F_T^\star + \sum_{k=2}^{T-1}\gamma^{T-k+1}F_k^\star + \frac{\gamma^T}{1-\gamma}F_1^\star$$

$$= \sum_{k=2}^{T}\gamma^{T-k+1}F_k^\star - \gamma^{T-1}F_1^\star.$$

Thus by taking expectation conditioned on $E^{\theta^\star}$ over the absolute value of RHS, we have

$$\mathbb{E}_{E^{\theta^\star}}\left[\sum_{t=1}^{T}\sum_{k=1}^{t-1}\gamma^{t-k}\left(F_{k+1}^\star - F_k^\star\right)\right] \le \mathbb{E}_{E^{\theta^\star}}\left[\sum_{k=2}^{T}\gamma^{T-k+1}|F_k^\star|\right] + \mathbb{E}_{E^{\theta^\star}}\left[\gamma^{T-1}|F_1^\star|\right]$$

$$\le L\sum_{k=2}^{T}\gamma^{T-k+1}\cdot\mathbb{E}_{E^{\theta^\star}}\left[\|\widetilde{\theta}_k\|\right] + L\gamma^{T-1}\cdot\mathbb{E}_{E^{\theta^\star}}\left[\|\widetilde{\theta}_1\|\right]$$

$$\le \sum_{k=0}^{T-1}\gamma^k\left(2\|\theta^\star\|L + 2\sqrt{\frac{d}{\lambda}}L\right)$$

$$\le \frac{1}{1-\gamma}\left(B_f^{\theta^\star} + 2\sqrt{\frac{d}{\lambda}}L\right). \qquad \text{(A.47)}$$

### A.8.6 Term (A.42)

We start off by rearranging terms in the summation: $\sum_{t=1}^{T}\sum_{k=1}^{t-1}\gamma^{t-k}\left(F_k(S_k) - F_{k+1}(S_k)\right)$.

$$\sum_{t=1}^{T}\sum_{k=1}^{t-1}\gamma^{t-k}\left(F_k(S_k) - F_{k+1}(S_k)\right) = \sum_{k=1}^{T-1}\left(F_k(S_k) - F_{k+1}(S_k)\right)\sum_{t=k+1}^{T}\gamma^{t-k}$$

$$= \sum_{k=1}^{T-1}\frac{\gamma - \gamma^{T-k+1}}{1-\gamma}\left(F_k(S_k) - F_{k+1}(S_k)\right)$$

$$= \sum_{k=1}^{T-1}\frac{\gamma - \gamma^{T-k+1}}{1-\gamma}\langle\widetilde{\theta}_k - \widetilde{\theta}_{k+1}, \frac{1}{M}\sum_{x\in S_k}x\rangle$$

$$\le \frac{1}{1-\gamma}\frac{1}{M}\sum_{t=1}^{T-1}\sum_{i=1}^{M}\left|\langle\widetilde{\theta}_t - \widetilde{\theta}_{t+1}, x_{t,i}\rangle\right|. \qquad \text{(A.48)}$$

In the following Corollary A.13, we bound the RHS above by constructing a high probability confidence ellipsoid for $\widetilde{\theta}_{t+1} - \widetilde{\theta}_t$ and then completing with a call of Lemma A.10.

**Corollary A.13.** For any realization of $\theta^\star$, conditioned on event $E^{\theta^\star}$, it holds that

$$\sum_{t=1}^{T}\sum_{k=1}^{t-1}\gamma^{t-k}\left(F_k(S_k) - F_{k+1}(S_k)\right) \leq \frac{1}{1-\gamma}\frac{1}{M}\operatorname{RGT}\left(2\beta_T^{\theta^\star}\left(\frac{\delta}{2}\right) + 2\alpha_T\left(\frac{\delta}{2}\right)\right). \quad \text{(A.49)}$$

*Proof.* See Appendix D.9. $\qquad\square$

Therefore, after taking expectation conditioned on $E^{\theta^\star}$, we still have

$$\mathbb{E}_{E^{\theta^\star}}\left[\sum_{t=1}^{T}\sum_{k=1}^{t-1}\gamma^{t-k}\left(F_k(S_k) - F_{k+1}(S_k)\right)\right] \leq \frac{1}{M}\cdot\frac{1}{1-\gamma}\operatorname{RGT}\left(2\beta_T^{\theta^\star}\left(\frac{\delta}{2}\right) + 2\alpha_T\left(\frac{\delta}{2}\right)\right). \tag{A.50}$$

### A.8.7 Pulling $4$ terms into the final bound of the first half (A.21)

Going back to the contribution coming from the first half of the regret decomposition (A.18), plugging (A.43), (A.46), (A.47) and (A.50) into (A.38), it holds that, for $\forall\delta\in(0,1)$

$$M\mathbb{E}_{\theta^\star}\left[\sum_{t=1}^{T}\min\left\{F_t^\star - F_t(S_t), B_f^{\theta^\star}\right\}\right] \leq 2\delta MT\cdot B_f^{\theta^\star} + (1-2\delta)M\cdot\mathbb{E}_{E^{\theta^\star}}\left[\sum_{t=1}^{T}F_t^\star - F_t(S_t)\right]$$

$$\leq 2\delta MT\cdot B_f^{\theta^\star} + (1-2\delta)\cdot\frac{3M}{1-\gamma}\left(B_f^{\theta^\star} + 2\sqrt{\frac{d}{\lambda}}L\right).$$

$$+ (1-2\delta)\cdot\frac{1}{1-\gamma}\operatorname{RGT}\left(2\beta_T^{\theta^\star}\left(\frac{\delta}{2}\right) + 2\alpha_T\left(\frac{\delta}{2}\right)\right). \tag{A.51}$$

Averaging (A.51) over the prior of $\theta^\star$, we have

$$M\mathbb{E}\left[\sum_{t=1}^{T}\min\left\{F_t^\star - F_t(S_t), B_f^{\theta^\star}\right\}\right] = \mathbb{E}_{\theta^\star\sim\pi}\left[M\mathbb{E}_{\theta^\star}\left[\sum_{t=1}^{T}\min\left\{F_t^\star - F_t(S_t), B_f^{\theta^\star}\right\}\right]\right]$$

$$\leq 2\delta\cdot\mathbb{E}_{\theta^\star\sim\pi}\left[B_f^{\theta^\star}\right]\cdot MT$$

$$+ \frac{3(1-2\delta)}{1-\gamma}\cdot\left(\mathbb{E}_{\theta^\star\sim\pi}\left[B_f^{\theta^\star}\right] + 2\sqrt{\frac{d}{\lambda}}L\right)\cdot M$$

$$+ \frac{1-2\delta}{1-\gamma}\cdot\mathbb{E}_{\theta^\star\sim\pi}\left[\operatorname{RGT}\left(2\beta_T^{\theta^\star}\left(\frac{\delta}{2}\right) + 2\alpha_T\left(\frac{\delta}{2}\right)\right)\right]. \tag{A.52}$$

### A.9 The Second Half of Regret Bound as in (A.22)

Conditioned on $E^{\theta^\star}\left(\frac{\delta}{2}\right)$, which holds with probability $1-\delta$, $\widetilde{\theta}_t$ lies in a confidence ellipsoid around $\theta^\star$ at all times,

$$\|\widetilde{\theta}_t - \theta^\star\|_{V_t} \leq \eta_t(\delta), \quad \forall t\in[T].$$

We wrap up an upper bound for $\sum_{t=1}^{T}\sum_{i=1}^{M}\left|\langle\widetilde{\theta}_t - \theta^\star, x_{t,i}\rangle\right|$ derived by calling Lemma A.10 into the corollary as follows.

**Corollary A.14.** Conditioned on $E^{\theta^\star}\left(\frac{\delta}{2}\right)$, the part of total regret contributed by the prediction error of TS sampled $\widetilde{\theta}_t$ is upper bounded by

$$\sum_{t=1}^{T}\sum_{i=1}^{M}\left|\langle\widetilde{\theta}_t - \theta^\star, x_{t,i}\rangle\right| \leq \operatorname{RGT}\left(\beta_T^{\theta^\star}\left(\frac{\delta}{2}\right) + \alpha_T\left(\frac{\delta}{2}\right)\right). \tag{A.53}$$

*Proof.* Lemma A.10 directly applies by customizing the parameter $\eta_t(\delta)$ to be $\beta_T^{\theta^\star}\left(\frac{\delta}{2}\right)+\alpha_T\left(\frac{\delta}{2}\right)$. □

With Corollary A.14 ready, we take expectation first conditioned on $\theta^\star$ and then over the prior of $\theta^\star$, which finally gives an upper bound of (A.22) as

$$\mathbb{E}_{\theta^\star\sim\pi}\left[\mathbb{E}_{\theta^\star}\left[\sum_{t=1}^{T}\sum_{i=1}^{M}\min\left\{\left|\langle\widetilde{\theta}_t-\theta^\star,x_{t,i}\rangle\right|,B_f^{\theta^\star}\right\}\right]\right]$$

$$\leq\mathbb{E}_{\theta^\star\sim\pi}\left[\delta MTB_f^{\theta^\star}+(1-\delta)\operatorname{RGT}\left(\beta_T^{\theta^\star}\left(\frac{\delta}{2}\right)+\alpha_T\left(\frac{\delta}{2}\right)\right)\right]$$

$$=\delta\cdot\mathbb{E}_{\theta^\star\sim\pi}\left[B_f^{\theta^\star}\right]\cdot MT+(1-\delta)\cdot\mathbb{E}_{\theta^\star\sim\pi}\left[\operatorname{RGT}\left(\beta_T^{\theta^\star}\left(\frac{\delta}{2}\right)+\alpha_T\left(\frac{\delta}{2}\right)\right)\right]. \tag{A.54}$$

## A.10 Final Bound: Combining The Two Halves (A.21) and (A.22)

Pulling two parts (A.52) and (A.54) into the regret decomposition (A.18), for $\forall\delta\in(0,1)$, with $\gamma$ s.t. $\frac{1}{1-\gamma}=O\left(\frac{\sqrt{d}}{\mu}\right)$, the Bayesian regret of Alg.1 is bounded by

$$\operatorname{BayesRGT}(T,M)\leq 3\delta\cdot\mathbb{E}_{\theta^\star\sim\pi}\left[B_f^{\theta^\star}\right]\cdot MT$$

$$+\frac{3(1-2\delta)}{1-\gamma}\cdot\left(\mathbb{E}_{\theta^\star\sim\pi}\left[B_f^{\theta^\star}\right]+2\sqrt{\frac{d}{\lambda}}L\right)\cdot M$$

$$+(1-\delta)\cdot\mathbb{E}_{\theta^\star\sim\pi}\left[\operatorname{RGT}\left(\beta_T^{\theta^\star}\left(\frac{\delta}{2}\right)+\alpha_T\left(\frac{\delta}{2}\right)\right)\right]$$

$$+\frac{1-2\delta}{1-\gamma}\cdot\mathbb{E}_{\theta^\star\sim\pi}\left[\operatorname{RGT}\left(2\beta_T^{\theta^\star}\left(\frac{\delta}{2}\right)+2\alpha_T\left(\frac{\delta}{2}\right)\right)\right].$$

By taking the probability of failure $\delta$ to be of $O(\frac{1}{T})$, we finally arrive at

$$\operatorname{BayesRGT}(T,M)\leq O\left(\frac{1}{1-\gamma}\cdot\left(\mathbb{E}_{\theta^\star\sim\pi}\left[B_f^{\theta^\star}\right]+2\sqrt{\frac{d}{\lambda}}L\right)\cdot M\right.$$

$$\left.+\frac{1}{1-\gamma}\cdot\mathbb{E}_{\theta^\star\sim\pi}\left[\operatorname{RGT}\left(2\beta_T^{\theta^\star}\left(\frac{1}{2T}\right)+2\alpha_T\left(\frac{1}{2T}\right)\right)\right]\right), \tag{A.55}$$

where $B_f^{\theta^\star}=2L\|\theta^\star\|$ and $\frac{1}{1-\gamma}=O\left(\frac{\sqrt{d}}{\mu}\right)$.

The orders of two expectations in (A.55) is claimed as follows.

**Claim A.15.** The orders of $\mathbb{E}_{\theta^\star\sim\pi}\left[B_f^{\theta^\star}\right]$ and $\mathbb{E}_{\theta^\star\sim\pi}\left[\operatorname{RGT}\left(2\beta_T^{\theta^\star}\left(\frac{1}{2T}\right)+2\alpha_T\left(\frac{1}{2T}\right)\right)\right]$ are:

- $\mathbb{E}_{\theta^\star\sim\pi}\left[B_f^{\theta^\star}\right]$ is of order

$$O\left(\sqrt{\frac{d}{\lambda}}L\right). \tag{A.56}$$

- $\mathbb{E}_{\theta^\star\sim\pi}\left[\operatorname{RGT}\left(2\beta_T^{\theta^\star}\left(\frac{1}{2T}\right)+2\alpha_T\left(\frac{1}{2T}\right)\right)\right]$ is of order

$$O\left(\frac{L}{\sqrt{\lambda}}d\sqrt{M}(\sqrt{T}+\sqrt{dM})\log\left(\frac{\sigma^2\lambda d+TML^2}{\sigma^2\lambda d}\right)\right). \tag{A.57}$$

*Proof.* See Appendix D.10. □

Therefore, use $\widetilde{O}$ to hide logarithmic term and lower $O(1)$ order term on $T$, recall $\frac{1}{1-\gamma}=O\left(\frac{\sqrt{d}}{\mu}\right)$ and $L=\sqrt{d}$, we finally arrived at a Bayesian regret of order

$$\operatorname{BayesRGT}(T,M)=\widetilde{O}\left(\frac{\sqrt{d}}{\mu}\cdot\frac{L}{\sqrt{\lambda}}\cdot d\sqrt{MT}\right)=\widetilde{O}\left(\frac{d}{\mu\sqrt{\lambda}}\cdot d\sqrt{MT}\right). \tag{A.58}$$

# B  Real-world experiment validation

## B.1  Optimizing sequence fitness for CRISPR gene-editing

Our TS-DE method was adapted for use in a gene-editing application in real-world experiments. Briefly, gene-editing, exemplified by technology derived from the Clustered Regularly Interspaced Short Palindromic Repeats, or CRISPR system, is a powerful tool for engineering genetic information in living organisms, and has transformed basic research and human therapeutics [13]. The efficiency and outcome of CRISPR gene-editing is highly dependent on the selection of guideRNA sequences, which form a complex with CRISPR proteins to perform gene-editing [44]. The TS-DE was applied to guide high-throughput CRISPR gene-editing experiments. In particular, we use known genomic motif features and a linear model for modeling the log editing capacity. At the beginning of each round of experiment, we computationally generate a new library of design sequences by randomly generating mutations and recombinations based on the previous population. Then we apply the bandit linear model to select sequences with high predicted fitness, and evaluate their actual editing capacities in the next round of experiments. A total of 14,358 unique guideRNA sequences were measured, and the log capacity improved by $\approx 5$. Notably, the optimized CRISPR designs generated by our DE approach is part of another manuscript [26], demonstrating real-world utility of current method. See Fig.B.1 for an illustration of the pipeline. We refer to Hughes et al, 2022 for more details on the experiment and computation.

**Remark**  The above real-world application of bandit DE differs from Algorithm 1 and generalizes it in a number of ways. For example, features used for predicting the gene-editing efficiency are not limited to motif features. Also they are not binary valued. Second, recombination and mutation were not done exactly as in Modules 1 and 2. They were randomized on the basepair level rather than the motif level. Despite these differences, our method was able to guide the experiment and accelerate discovery. This demonstrates the bandit DE method may have broad generalizability and it is not restricted to the abstract mathematical model formulated in this paper.

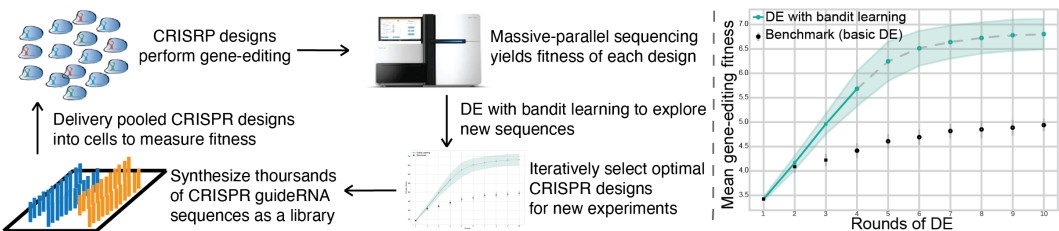

Figure B.1: **Evolving CRISPR sequences using iterative real-world experiments and accelerated DE** Left panels: Workflow overview. Right panel: Fitness distribution showing accelerated optimization using DE with Bandit learning. (Ths figure is borrowed from [26])

# C  Proof of Theorem 5.1

## C.1  Ascent property of `Crossover_Selection`

*Proof.* Since each $z \in S'$ is generated in the same way independently and $F(S')$ is the fitness averaging over all $z$'s, thus

$$\mathbb{E}\left[F(S')\right] = \mathbb{E}\left[f(z)\,|\,z \in S'\right],$$

with the expectation taken over the randomness in sampling $z$'s parents $x$ and $y$ and in crossing over $x$ and $y$. Using notation $\mathbb{E}_{x,y}\left[\cdot\right] := \mathbb{E}\left[\cdot\,|\,x,y\right]$, the conditional expectation given $x$ and $y$, rewrite $\mathbb{E}\left[f(z)\,|\,z \in S'\right]$ as

$$\mathbb{E}\left[f(z)\,|\,z \in S'\right] = \mathbb{E}\left[\mathbb{E}_{x,y}\left[f(z)\,\middle|\,f(z) \geq \frac{f(x)+f(y)}{2}\right]\right],$$

where the inner expectation is taken over the randomness in the recombination step $z \leftarrow \mathtt{Rcb}(x,y)$, and the outer expectation is over sampling $x$ and $y$.

Given $x$ and $y$, a recombined child sample $z$ can be represented by

$$z = \frac{x+y}{2} + \frac{x-y}{2} \cdot e, \tag{C.1}$$

where the $\cdot$ operator here multiplies two vector entrywisely into a new vector and $e$ is a vector consisting of $d$ independent Rademacher variables, that is $e = (e_i, \cdots, e_d)^\top$ and

$$e_i \overset{\text{i.i.d.}}{\sim} \text{Rad}.$$

Thus $f(z)$ is computed as

$$f(z) = \frac{f(x)+f(y)}{2} + \frac{1}{2}\sum_{i=1}^{d} \theta_i (x_i - y_i) e_i. \tag{C.2}$$

And then $f(z) \geq \frac{f(x)+f(y)}{2}$ is equivalent to $\sum_{i=1}^{d} \theta_i (x_i - y_i) e_i \geq 0$, so

$$\mathbb{E}_{x,y}\left[ f(z) \,\middle|\, f(z) \geq \frac{f(x)+f(y)}{2} \right]$$

$$= \frac{f(x)+f(y)}{2} + \frac{1}{2}\mathbb{E}\left[ \sum_{i=1}^{d} \theta_i (x_i - y_i) e_i \,\middle|\, \sum_{i=1}^{d} \theta_i (x_i - y_i) \geq 0 \right]$$

$$\frac{f(x)+f(y)}{2} + \frac{1}{2}\mathbb{E}\left[ \left| \sum_{i=1}^{d} \theta_i (x_i - y_i) e_i \right| \right] \tag{C.3}$$

$$\geq \frac{f(x)+f(y)}{2} + \frac{C}{2}\| \theta \cdot (x-y) \|,$$

where (C.3) holds because $\sum_{i=1}^{d} \theta_i (x_i - y_i) e_i$ is symmetrically distributed around $0$. And in the last line, $\cdot$ is still the entrywise multiplication between vectors and $C \geq \frac{1}{\sqrt{2}}$ according to Haagerup [20].

Thus,

$$\mathbb{E}\left[ F(S') \right] = \mathbb{E}\left[ \mathbb{E}_{a,b}\left[ f(z) \,\middle|\, f(z) \geq \frac{f(x)+f(y)}{2} \right] \right]$$

$$\geq \mathbb{E}\left[ \frac{f(x)+f(y)}{2} \right] + \frac{1}{2\sqrt{2}}\mathbb{E}\left[ \| \theta \cdot (x-y) \| \right]$$

$$\geq F(S) + \frac{1}{2\sqrt{2}}\mathbb{E}\left[ \| \theta \cdot (x-y) \| \right]. \tag{C.4}$$

By Cauchy-Schwarz, we have

$$\| \theta \cdot (x-y) \| \geq \frac{1}{\sqrt{d}}\sum_{i=1}^{d} |\theta_i| |x_i - y_i|$$

Thus, by averaging over all $x$ and $y$ sampled from $S$,

$$\mathbb{E}\left[ \| \theta \cdot (x-y) \| \right] \geq \frac{1}{\sqrt{d}}\sum_{i=1}^{d} |\theta_i| \mathbb{E}\left[ |x_i - y_i| \right]$$

When $\forall i \in [d], x_i, y_i \in \{0,1\}$ for all $x$ and $y$'s in $S$, then

$$\mathbb{E}\left[ |x_i - y_i| \right] \geq \mathbb{E}\left[ (x_i - y_i)^2 \right] = 2Var_i(S),$$

$$\mathbb{E}\left[ \| \theta \cdot (x-y) \| \right] \geq \frac{2}{\sqrt{d}}\sum_{i=1}^{d} |\theta_i| Var_i(S), \tag{C.5}$$

where $Var_i(S)$ denotes the variance of $x_i$ when $x$ is uniformly sampled from $S$.

Therefore,

$$\mathbb{E}\left[ F(S') \right] \geq F(S) + \frac{1}{\sqrt{2d}}\sum_{i} |\theta_i| Var_i(S).$$

$$\square$$

# D Omitted Proofs in Appendix A

## D.1 Proof of Lemma A.4

*Proof.* For $\forall i \notin \mathcal{I}, \forall x \in S$ is not induced to mutate at site $i$, thus for $x' = \text{Mut}(x, \mathcal{I}, \mu)$, $x'_i = x_i$ and
$$p_i(S') = p_i(S).$$

For $i \in \mathcal{I}$, after the directed mutation formulated as (3.3), $\mathbb{E}\left[\mathbf{I}\{x'_i = 1\}\right] = (1 - \mu)\mathbf{I}\{x_i = 1\} + \frac{\mu}{2}$.
$$\mathbb{E}\left[p_i(S')\right] = (1 - \mu)p_i(S) + \frac{\mu}{2} = p_i(S) + \left(\frac{1}{2} - p_i(S)\right)\mu.$$

Since $i \in \mathcal{I}$ iff $\frac{1}{M}\sum_{x \in S}\theta_i \cdot x_i \leq \theta_i \cdot \bar{x}_i$, which is equivalent to $p_i(S) \leq \frac{1}{2}$, showing that the $i$-th dimension is not sufficient with the favored feature. Then the directed mutation strictly increases $p_i(S)$ for any insufficient dimension $i$ by boosting it by $\mu\left(\frac{1}{2} - p_i(S)\right) \geq 0$, which resulting in a $|\theta_i|$-increase in the utility value per unit of increase in $p_i(S)$.

Therefore, $\mathbb{E}\left[F(S')\right] \geq F(S)$ and
$$\mathbb{E}\left[p_i(S')\right] = p_i(S) > \frac{1}{2}, \quad \forall i \notin \mathcal{I},$$
$$\mathbb{E}\left[p_i(S')\right] = p_i(S) + \left(\frac{1}{2} - p_i(S)\right)\mu \geq \frac{\mu}{2}, \quad \forall i \in \mathcal{I}.$$

Thus, after calling $S' = \text{Directed\_Mutation}(f, S, \mu)$, $\mathbb{E}\left[p_i(S')\right] \geq \frac{\mu}{2}, \forall i \in [d]$. By a standard argument of concentration and a union bound taken over $i \in [d]$, with probability $1 - \delta$,
$$p_i(S') \geq \frac{\mu}{4}, \quad \forall i \in [d]$$
when $|S| = \Omega\left(\frac{\log(\frac{d}{\delta})}{\mu^2}\right)$. $\qquad\square$

## D.2 Proof of Lemma A.6

*Proof.* Lemma A.6 is derived by taking union bound over $t + 1 \in [T]$ upon $\min_i p_i^{\widetilde{\theta}_{t+1}}(S'_t) \geq \frac{\mu}{4}$ obtained by instantiating (A.4) for $S'_t$ and $f_{\widetilde{\theta}_{t+1}}$ in Lemma A.4. $\qquad\square$

## D.3 Proof of Lemma A.7

*Proof.* Recall from Theorem 5.1 that
$$\mathbb{E}\left[F(S')\right] \geq F(S) + \frac{1}{\sqrt{2d}}\sum_i |\theta_i|\text{Var}_i(S), \tag{5.1 revisited}$$

where $Var_i(S)$ is the variance of $x_i$ when $x$ is uniformly sampled from $S$. Using $p_i(S)$ defined in Definition A.3
$$\text{Var}_i(S) = p_i(S)\left(1 - p_i(S)\right). \tag{D.1}$$

Then, it suffices to prove
$$\sum_{i=1}^{d} |\theta_i| p_i(S)\left(1 - p_i(S)\right) \geq \min_i p_i(S) \cdot \left(F^\star - F(S)\right).$$

Taking a closer look at the suboptimality gap $F^\star - F(S)$, it is easily observed that
$$F^\star = \sum_{i:\theta_i \geq 0} \theta_i + \sum_{i:\theta_i < 0} 0, \tag{D.2}$$
$$F(S) = \sum_{i:\theta_i \geq 0} \theta_i \left[p_i(S) \cdot 1 + (1 - p_i(S)) \cdot 0\right] + \sum_{i:\theta_i < 0} \theta_i \left[p_i(S) \cdot 0 + (1 - p_i(S)) \cdot 1\right]$$
$$= \sum_{i:\theta_i \geq 0} \theta_i \cdot p_i(S) + \sum_{i:\theta_i < 0} \theta_i \cdot (1 - p_i(S)). \tag{D.3}$$

Plugging in (D.2) and (D.3), we have

$$F^\star - F(S) = \sum_i |\theta_i|(1 - p_i(S)). \tag{D.4}$$

Therefore,

$$\sum_{i=1}^{d} |\theta_i| p_i(S) (1 - p_i(S)) \geq \min_i p_i(S) \cdot (F^\star - F(S)).$$

$\square$

## D.4 Proof of Lemma A.9

*Proof.* We finish the proof by lower bounding the probabilities of two events $\widehat{E}^{\theta^\star}\left(\frac{\delta}{2}\right)$ and $\widetilde{E}^{\theta^\star}\left(\frac{\delta}{2}\right)$ by $1 - \frac{\delta}{2}$ separately. Recall that for $\forall t \in [T]$

$$\widetilde{\theta}_t \sim \mathcal{N}(\widehat{\theta}_t, V_t^{-1}), \tag{A.9 revisited}$$

$$V_t = \frac{1}{\sigma^2}\Phi_{t-1}^\top \Phi_{t-1} + \lambda I, \tag{A.10 revisited}$$

$$\widehat{\theta}_t = \frac{1}{\sigma^2} V_t^{-1} \Phi_{t-1}^\top U_{t-1}. \tag{A.11 revisited}$$

**Bounding $\mathbb{P}\left(\widehat{E}^{\theta^\star}\left(\frac{\delta}{2}\right)\right)$.**

Plugging (A.10) into (A.11), we will see $\widehat{\theta}_t$ is related to the regularized least square estimator (RLS):

$$\widehat{\theta}_t = \frac{1}{\sigma^2} V_t^{-1} \Phi_{t-1}^\top U_{t-1} = \left(\Phi_{t-1}^\top \Phi_{t-1} + \sigma^2 \lambda I\right)^{-1} \Phi_{t-1}^\top U_{t-1}.$$

For any fixed ground truth $\theta^\star$, $\widehat{\theta}_t$ is a RLS estimator of $\theta^\star$ regularized by $\sigma^2 \lambda \cdot I$. Conditioned on $\theta^\star$, define a filtration w.r.t. the data $\{(x_{t,i}, u(x_{t,i})), i \in [M], t \in [T-1]\}$ collected along the way.

**Definition D.1.** Define $\mathcal{F}_t$ be the information accumulated after the $t$-th batch of data points is collected.

$$\mathcal{F}_0 := \sigma(\theta^\star) \tag{D.5}$$
$$\mathcal{F}_t := \{\mathcal{F}_{t-1}, \sigma(x_{t,1}, u(x_{t,1}), \cdots, x_{t,M}, u(x_{t,M}))\}. \tag{D.6}$$

Then we fine grind the filtration $\{F_t\}_{t=0}^{T-1}$ to be

$$\mathcal{F}_0 \subset \mathcal{F}_{0,1} \subset \cdots \subset \mathcal{F}_{0,M} \subset \mathcal{F}_1 \subset \cdots \subset \mathcal{F}_{t-1} \subset \mathcal{F}_{t-1,1} \subset \cdots \mathcal{F}_{t-1,M} \subset \mathcal{F}_{t-1} \subset \cdots \subset \mathcal{F}_{T-1} \tag{D.7}$$

by essentially adding $M$ layers between $\mathcal{F}_{t-1}$ and $\mathcal{F}_t$ and each layer $\mathcal{F}_{t-1,i}$ contains the information obtained after $(x_{t,i}, u(x_{t,i}))$ is added to the dataset.

Under Assumption 3.5, each feedback $u(x_{t,i})$ satisfies

$$u(x_{t,i}) = f_{\theta^\star}(x_{t,i}) + \xi_{t,i}, \tag{A.8 revisited}$$

where

$$\xi_{t,i} | \mathcal{F}_{t,i} \sim \mathcal{N}\left(0, \sigma^2\right).$$

Then bounding $\mathbb{P}\left(\widehat{E}^{\theta^\star}\left(\frac{\delta}{2}\right)\right)$ is a straightforward application of the Theorem 2 in [1], wrapped up into the following proposition.

**Proposition D.2.** Under Assumption 3.5, for $\forall \delta \in (0,1)$ and any $\mathcal{F}_{t,i}$-adapted data sequence $\left(\{x_{0,i}\}_{i=1}^M, \cdots, \{x_{T-1,i}\}_{i=1}^M\right)$ s.t. $\|x_{t,i}\| \leq L$,

$$\mathbb{P}\left(\exists t \in [T] : \left\|\widehat{\theta}_t - \theta^\star\right\|_{V_t} \geq \beta_t(\delta) \bigg| \mathcal{F}_0\right) \leq \delta. \tag{D.8}$$

From the result above, we have

$$\mathbb{P}\left(\widehat{E}^{\theta^\star}\left(\frac{\delta}{2}\right)\right) = \mathbb{P}\left(\left\|\widehat{\theta}_t - \theta^\star\right\|_{V_t} \le \beta_t\left(\frac{\delta}{2}\right), \forall t \in [T] \,\middle|\, \theta^\star\right)$$

$$= 1 - \mathbb{P}\left(\exists t \in [T]: \left\|\widehat{\theta}_t - \theta^\star\right\|_{V_t} \ge \beta_t\left(\frac{\delta}{2}\right) \,\middle|\, \theta^\star\right)$$

$$\ge 1 - \frac{\delta}{2}.$$

**Bounding** $\mathbb{P}\left(\widehat{E}^{\theta^\star}\left(\frac{\delta}{2}\right)\right)$**.** Recall that $\widetilde{\theta}_t$ is sampled from posterior distribution $\mathcal{N}(\widehat{\theta}_t, V_t^{-1})$ independently from $\theta^\star$, then we have

$$\left\|\widetilde{\theta}_t - \widehat{\theta}_t\right\|_{V_t}^2 = \left\|V_t^{\frac{1}{2}}(\widetilde{\theta}_t - \widehat{\theta}_t)\right\|^2, \quad \forall t \in [T] \tag{D.9}$$

where $V_t^{\frac{1}{2}}(\widetilde{\theta}_t - \widehat{\theta}_t) \sim \mathcal{N}(0, \mathbf{I})$. Thus $\left\|\widetilde{\theta}_t - \widehat{\theta}_t\right\|_{V_t}^2 \sim \chi_d^2$ independently from $\theta^\star$. From the concentration of $\chi_d^2$ random variable, we have

$$\mathbb{P}\left(\chi_d \ge 2\sqrt{d\log\left(\frac{1}{\delta}\right)} + \sqrt{d}\right) \le \delta.$$

Therefore, by taking a union bound over $\forall t \in [T]$, we have

$$\mathbb{P}\left(\widetilde{E}\left(\frac{\delta}{2}\right)\,\middle|\,\theta^\star\right) = \mathbb{P}\left(\widetilde{E}\left(\frac{\delta}{2}\right)\right)$$

$$= \mathbb{P}\left(\left\|\widetilde{\theta}_t - \widehat{\theta}_t\right\|_{V_t} \le \alpha_t\left(\frac{\delta}{2}\right), \forall t \in [T]\right)$$

$$\ge 1 - \sum_{t=1}^{T} \mathbb{P}\left(\left\|\widetilde{\theta}_t - \widehat{\theta}_t\right\|_{V_t} \le \alpha_t\left(\frac{\delta}{2}\right)\right)$$

$$\ge 1 - \sum_{t=1}^{T}\frac{\delta}{2T} = 1 - \frac{\delta}{2}.$$

$\square$

### D.5 Proof of Proposition D.2

*Proof.* Use notation $\widetilde{V}_t := \Phi_{t-1}^\top\Phi_{t-1} + \sigma^2\lambda I$, then $V_t = \frac{1}{\sigma^2}\widetilde{V}_t$ and $\widehat{\theta}_t = \widetilde{V}_t^{-1}\Phi_{t-1}^\top U_{t-1}$. According to Theorem 2 in [1], for $\forall \delta \in (0,1)$ and any $\mathcal{F}_{t,i}$-adapted data sequence $\left(\{x_{0,i}\}_{i=1}^{M}, \cdots, \{x_{T-1,i}\}_{i=1}^{M}\right)$ s.t. $\|x_{t,i}\| \le L$,

$$\mathbb{P}\left(\exists t \in [T]: \left\|\widehat{\theta}_t - \theta^\star\right\|_{\widetilde{V}_t} \ge \sigma \cdot \beta_t(\delta)\,\middle|\,\mathcal{F}_0\right) \le \delta.$$

Therefore, since $\left\|\widehat{\theta}_t - \theta^\star\right\|_{V_t} = \left\|\widehat{\theta}_t - \theta^\star\right\|_{\frac{1}{\sigma^2}\widetilde{V}_t} = \frac{1}{\sigma}\left\|\widehat{\theta}_t - \theta^\star\right\|_{\widetilde{V}_t}$

$$\mathbb{P}\left(\exists t \in [T]: \left\|\widehat{\theta}_t - \theta^\star\right\|_{V_t} \ge \beta_t(\delta)\,\middle|\,\mathcal{F}_0\right) \le \delta.$$

$\square$

### D.6 Proof of Lemma A.10

*Proof.* We are about to take a closer look at the incremental increase of the determinant of $V_t$, define $V_{t,l} = \frac{1}{\sigma^2}\left(\sigma^2\lambda I + \sum_{i=1}^{t-1}\sum_{j=1}^{M}x_{i,j}x_{i,j}^T + \sum_{j=1}^{l}x_{t,j}x_{t,j}^T\right)$ for $\forall t \in [T], l \in [M]$ and thus

$V_{t,0} = V_t$. Mark the time steps where $V_t$ has significant increase in its determinant by $C := \{t \in [T] : \frac{\det(V_{t+1})}{\det(V_t)} > 2\}$. Then the prediction errors in $T$ rounds can be divided into two parts as

$$\sum_{t=1}^{T}\sum_{i=1}^{M} |\langle a_t - b_t, x_{t,i}\rangle| = \sum_{t\notin C}\sum_{i=1}^{M} |\langle a_t - b_t, x_{t,i}\rangle| + \sum_{t\in C}\sum_{i=1}^{M} |\langle a_t - b_t, x_{t,i}\rangle|. \tag{D.10}$$

The first half of (D.10) consists of error accumulated in the rounds where $\det(V_t)$ didn't increased much after having a batch update of size $M$, so we bound this part in the same spirit of bounding the case where only rank-1 update happens per round. Result is stated in the following claim.

**Claim D.3.** The first half of (D.10) is bounded by

$$\sum_{t\notin C}\sum_{i=1}^{M} |\langle a_t - b_t, x_{t,i}\rangle| \le \eta_T(\delta)\sqrt{\frac{2L^2 + 2\lambda}{\lambda}}\sqrt{MT\log\left(\frac{\det(V_{T+1})}{\det(V_1)}\right)}. \tag{D.11}$$

For the second half of (D.10), we are about to bound by showing $|C|$ is small. Notice that

$$\frac{\det(V_{T+1})}{\det(V_1)} = \prod_{t=1}^{T}\frac{\det(V_{t+1})}{\det(V_t)} \ge \prod_{t\in C}\frac{\det(V_{t+1})}{\det(V_t)} \ge 2^{|C|}, \tag{D.12}$$

thus $|C|$ should not be greater than $2\log\left(\frac{\det(V_{T+1})}{\det(V_1)}\right)$. And for $\forall t \in [T], i \in [M]$

$$|\langle a_t - b_t, x_{t,i}\rangle| \le \|a_t - b_t\|_{V_t}\|x_{t,i}\|_{V_t^{-1}} \le \frac{\eta_T(\delta)L}{\sqrt{\lambda}}. \tag{D.13}$$

Putting two parts together, we have

$$\sum_{t=1}^{T}\sum_{i=1}^{M} |\langle a_t - b_t, x_{t,i}\rangle| = \sum_{t\notin C}\sum_{i=1}^{M} |\langle a_t - b_t, x_{t,i}\rangle| + \sum_{t\in C}\sum_{i=1}^{M} |\langle a_t - b_t, x_{t,i}\rangle|$$

$$\le \eta_T(\delta)\sqrt{\frac{2L^2 + 2\lambda}{\lambda}}\sqrt{MT\log\left(\frac{\det(V_{T+1})}{\det(V_1)}\right)} + \eta_T(\delta)\frac{L}{\sqrt{\lambda}}|C|M$$

$$\le \eta_T(\delta)\sqrt{\frac{2L^2 + 2\lambda}{\lambda}}\sqrt{MT\log\left(\frac{\det(V_{T+1})}{\det(V_1)}\right)} + \eta_T(\delta)\frac{2L}{\sqrt{\lambda}}M\log\left(\frac{\det(V_{T+1})}{\det(V_1)}\right)$$

$$\le \eta_T(\delta)\sqrt{\frac{2L^2 + 2\lambda}{\lambda}}\sqrt{dMT\log\left(\frac{\sigma^2 d\lambda + MTL^2}{\sigma^2 d\lambda}\right)} + \eta_T(\delta)\frac{2L}{\sqrt{\lambda}}dM\log\left(\frac{\sigma^2 d\lambda + MTL^2}{\sigma^2 d\lambda}\right).$$

where the final line is referring to the result in [1] that

$$\log\left(\frac{\det(V_{T+1})}{\det(V_1)}\right) \le d\log\left(\frac{\sigma^2 d\lambda + MTL^2}{\sigma^2 d\lambda}\right).$$

$\square$

### D.7 Proof of Claim D.3

*Proof.* With probability $1 - \delta$, for $\forall t \in [T]$, normalized by $V_t$, $a_t$ and $b_t$ concentrate around each other with in a radius of $\eta_t(\delta)$, thus

$$|\langle a_t - b_t, x_{t,i}\rangle| \le \|a_t - b_t\|_{V_t}\|x_{t,i}\|_{V_t^{-1}} \le \eta_t(\delta)\|x_{t,i}\|_{V_t^{-1}}.$$

Summing over all individuals in time steps $t \notin C$, we have

$$\sum_{t \notin C} \sum_{i=1}^{M} |\langle a_t - b_t, x_{t,i} \rangle| \leq \sum_{t \notin C} \sum_{i=1}^{M} \eta_t(\delta) \|x_{t,i}\|_{V_t^{-1}}$$

$$\leq \eta_T(\delta) \sum_{t \notin C} \sum_{i=1}^{M} \|x_{t,i}\|_{V_t^{-1}}$$

$$\leq \eta_T(\delta) \sqrt{MT \sum_{t \notin C} \sum_{i=1}^{M} \|x_{t,i}\|_{V_t^{-1}}^2}$$

$$\leq \eta_T(\delta) \sqrt{\frac{L^2 + \lambda}{\lambda} \cdot MT \sum_{t \notin C} \sum_{i=1}^{M} \log \left(1 + \|x_{t,i}\|_{V_t^{-1}}^2 \right)}, \qquad \text{(D.14)}$$

where (D.14) holds because

$$\|x_{t,i}\|_{V_t^{-1}}^2 \leq \lambda_{\max}(V_t^{-1}) \|x_{t,i}\|^2 \leq \frac{L^2}{\lambda}.$$

Continuing from (D.14), we can substitute the normalization matrix $V_t^{-1}$ with $V_{t,i}^{-1}$, at the cost of inflating by 2, and then following the classic self-normalized bound on data points. Recall the Lemma 12 in [1]:

$$\frac{\|x\|_A^2}{\|x\|_B^2} \leq \frac{\det(A)}{\det(B)}, \quad \text{if } A \succeq B. \qquad \text{(D.15)}$$

Substituting $V_t^{-1}$ with $V_{t,i-1}^{-1}$ in $\|x_{t,i}\|_{V_t^{-1}}^2$, noticing $\frac{\det(V_t^{-1})}{\det(V_{t,i-1}^{-1})} = \frac{\det(V_{t,i-1})}{\det(V_t)} \leq \frac{\det(V_{t+1})}{\det(V_t)} \leq 2$ when $t \notin C$, leads to

$$\|x_{t,i}\|_{V_t^{-1}}^2 \leq 2\|x_{t,i}\|_{V_{t,i}^{-1}}^2,$$

$$\log \left(1 + \|x_{t,i}\|_{V_t^{-1}}^2 \right) \leq \log \left(1 + 2\|x_{t,i}\|_{V_{t,i-1}^{-1}}^2 \right)$$

$$\leq 2 \log \left(1 + \|x_{t,i}\|_{V_{t,i-1}^{-1}}^2 \right).$$

Then it follows the self-normalized bound in [1] and gives that

$$\sum_{t \notin C} \sum_{i=1}^{M} \log(1 + \|x_{t,i}\|_{V_t^{-1}}^2) \leq 2 \sum_{t \notin C} \sum_{i=1}^{M} \log \left(1 + \|x_{t,i}\|_{V_{t,i-1}^{-1}}^2 \right)$$

$$\leq 2 \sum_{t=1}^{T} \sum_{i=1}^{M} \log \left(1 + \|x_{t,i}\|_{V_{t,i-1}^{-1}}^2 \right)$$

$$\leq 2 \log \left( \frac{\det(V_{T+1})}{\det(V_1)} \right).$$

Therefore, the first half of (D.10) is bounded by

$$\sum_{t \notin C} \sum_{i=1}^{M} |\langle a_t - b_t, x_{t,i} \rangle| \leq \eta_T(\delta) \sqrt{\frac{2L^2 + 2\lambda}{\lambda}} \sqrt{MT \log \left( \frac{\det(V_{T+1})}{\det(V_1)} \right)}.$$

$\square$

## D.8 Proof of Lemma A.12

*Proof.* By triangle inequality

$$\mathbb{E}_{E^{\theta^\star}} \left[ \|\widetilde{\theta}_t\| \right] \leq \|\theta^\star\| + \mathbb{E}_{E^{\theta^\star}} \left[ \|\widetilde{\theta}_t - \widehat{\theta}_t\| \right] + \mathbb{E}_{E^{\theta^\star}} \left[ \|\widehat{\theta}_t - \theta^\star\| \right].$$

Along with $\lambda_{\min}(V_t) \geq \lambda$ since $V_t$ is regularized with $\lambda I$ in its definition, then we have

$$\mathbb{E}_{E^{\theta^\star}}\left[\|\widetilde{\theta}_t\|\right] \leq \|\theta^\star\| + \frac{1}{\sqrt{\lambda}}\mathbb{E}_{E^{\theta^\star}}\left[\|\widetilde{\theta}_t - \widehat{\theta}_t\|_{V_t}\right] + \frac{1}{\sqrt{\lambda}}\mathbb{E}_{E^{\theta^\star}}\left[\|\widehat{\theta}_t - \theta^\star\|_{V_t}\right].$$

And in event $E^{\theta^\star} := E^{\theta^\star}\left(\frac{\delta}{2}\right) \cap E^{\theta^\star}_{\mathrm{DM}}$, $E^{\theta^\star}_{\mathrm{DM}}$ is independent from the sampling of $\widetilde{\theta}_t$, and conditioned on event $E^{\theta^\star}\left(\frac{\delta}{2}\right)$, both $\|\widetilde{\theta}_t - \widehat{\theta}_t\|_{V_t}$ and $\|\widehat{\theta}_t - \theta^\star\|_{V_t}$ tend to be smaller than it is unconditionally. Thus, we lift the condition on $E^{\theta^\star}$ to get an upper bound as

$$\mathbb{E}_{E^{\theta^\star}}\left[\|\widetilde{\theta}_t\|\right] \leq \|\theta^\star\| + \frac{1}{\sqrt{\lambda}}\mathbb{E}_{\theta^\star}\left[\|\widetilde{\theta}_t - \widehat{\theta}_t\|_{V_t}\right] + \frac{1}{\sqrt{\lambda}}\mathbb{E}_{\theta^\star}\left[\|\widehat{\theta}_t - \theta^\star\|_{V_t}\right]. \tag{D.16}$$

Recall that conditioned on any realization of $\theta^\star$, $\widetilde{\theta}_t$ is sampled from

$$\widetilde{\theta}_t \sim \mathcal{N}(\widehat{\theta}_t, V_t^{-1}) \tag{A.9 revisited}$$

with

$$V_t = \frac{1}{\sigma^2}\Phi_{t-1}^\top\Phi_{t-1} + \lambda I, \tag{A.10 revisited}$$

$$\widehat{\theta}_t = \frac{1}{\sigma^2}V_t^{-1}\Phi_{t-1}^\top U_{t-1}. \tag{A.11 revisited}$$

So $\left\|\widetilde{\theta}_t - \widehat{\theta}_t\right\|_{V_t}^2 \sim \chi_d^2$ independent from $\theta^\star$ and thus

$$\mathbb{E}_{\theta^\star}\left[\|\widetilde{\theta}_t - \widehat{\theta}_t\|_{V_t}\right] \leq \sqrt{d}. \tag{D.17}$$

Also, from (A.11), let $U_{t-1} = \Phi_{t-1}\theta^\star + \xi_{t-1}$ and $\xi_{t-1}$ be the corresponding noise vector, then $\widehat{\theta}_t - \theta^\star$ is computed as

$$\begin{aligned}
\widehat{\theta}_t - \theta^\star &= \frac{1}{\sigma^2}V_t^{-1}\Phi_{t-1}^\top U_{t-1} - \theta^\star \\
&\overset{A.10}{=} \left(\Phi_{t-1}^\top\Phi_{t-1} + \sigma^2\lambda I\right)^{-1}\Phi_{t-1}^\top U_{t-1} - \theta^\star \\
&= \left(\Phi_{t-1}^\top\Phi_{t-1} + \sigma^2\lambda I\right)^{-1}\Phi_{t-1}^\top(\Phi_{t-1}\theta^\star + \xi_{t-1}) - \theta^\star \\
&= \left(\Phi_{t-1}^\top\Phi_{t-1} + \sigma^2\lambda I\right)^{-1}\Phi_{t-1}^\top\xi_{t-1} - \sigma^2\lambda\left(\Phi_{t-1}^\top\Phi_{t-1} + \sigma^2\lambda I\right)^{-1}\theta^\star \\
&= \frac{1}{\sigma^2}V_t^{-1}\Phi_{t-1}^\top\xi_{t-1} - \lambda V_t^{-1}\theta^\star.
\end{aligned}$$

Thus,

$$\mathbb{E}_{\theta^\star}\left[\|\widehat{\theta}_t - \theta^\star\|_{V_t}\right] \leq \frac{1}{\sigma^2}\mathbb{E}_{\theta^\star}\left[\|V_t^{-1}\Phi_{t-1}^\top\xi_{t-1}\|_{V_t}\right] + \lambda\mathbb{E}_{\theta^\star}\left[\|V_t^{-1}\theta^\star\|_{V_t}\right], \tag{D.18}$$

where

$$\mathbb{E}_{\theta^\star}\left[\|V_t^{-1}\theta^\star\|_{V_t}\right] = \mathbb{E}_{\theta^\star}\left[\sqrt{\theta^{\star\top}V_t^{-1}\theta^\star}\right] \leq \frac{1}{\sqrt{\lambda}}\|\theta^\star\|, \tag{D.19}$$

and

$$\begin{aligned}
\mathbb{E}_{\theta^\star}\left[\|V_t^{-1}\Phi_{t-1}^\top\xi_{t-1}\|_{V_t}\right] &= \mathbb{E}_{\theta^\star}\left[\sqrt{\xi_{t-1}^\top\Phi_{t-1}V_t^{-1}\Phi_{t-1}^\top\xi_{t-1}}\right] \\
&= \mathbb{E}_{\theta^\star}\left[\mathbb{E}\left[\left.\sqrt{\xi_{t-1}^\top\Phi_{t-1}V_t^{-1}\Phi_{t-1}^\top\xi_{t-1}}\right| \Phi_{t-1}\right]\right] \\
&\overset{v:=\Phi_{t-1}^\top\xi_{t-1}}{=} \mathbb{E}_{\theta^\star}\left[\mathbb{E}\left[\left.\sqrt{v^\top V_t^{-1}v}\right| \Phi_{t-1}\right]\right],
\end{aligned}$$

with $v \in \mathbb{R}^d$ following the distribution $\mathcal{N}(0, \sigma^2\Phi_{t-1}^\top\Phi_{t-1})$ conditioned on $\Phi_{t-1}$ because the noise vector $\xi_{t-1} \mid \Phi_{t-1} \sim \mathcal{N}(0, \sigma^2 I)$. Recall $V_t = \frac{1}{\sigma^2}\Phi_{t-1}^\top\Phi_{t-1} + \lambda I$, therefore

$$\mathbb{E}_{\theta^\star}\left[\|V_t^{-1}\Phi_{t-1}^\top\xi_{t-1}\|_{V_t}\right] \leq \sigma^2\sqrt{d}. \tag{D.20}$$

Plugging (D.19) and (D.20) into (D.18), we have

$$\mathbb{E}_{\theta^\star} \left[ \|\widehat{\theta}_t - \theta^\star\|_{V_t} \right] \leq \sqrt{d} + \sqrt{\lambda}\|\theta^\star\|. \tag{D.21}$$

Then plug the inequality above together with (D.17) into (D.16), we finally arrive at

$$\mathbb{E}_{E^{\theta^\star}} \left[ \|\widetilde{\theta}_t\| \right] \leq 2\|\theta^\star\| + 2\sqrt{\frac{d}{\lambda}}.$$

$\square$

## D.9   Proof of Corollary A.13

*Proof.* As shown in Lemma A.9, conditioned on event $E^{\theta^\star}\left(\frac{\delta}{2}\right)$, TS estimate $\widetilde{\theta}_t$s are not far away from $\theta^\star$ simultaneously:

$$\|\widetilde{\theta}_t - \theta^\star\|_{V_t} \leq \beta_t^{\theta^\star}\left(\frac{\delta}{2}\right) + \alpha_t\left(\frac{\delta}{2}\right), \quad \forall t \in [T]. \tag{D.22}$$

Thus for $\forall t \in [T-1]$, $\widetilde{\theta}_t$ should not be far away from $\widetilde{\theta}_{t-1}$ with the same high probability. From equation (D.22), we have

$$\|\widetilde{\theta}_t - \theta^\star\|_{V_t} \leq \beta_t^{\theta^\star}\left(\frac{\delta}{2}\right) + \alpha_t\left(\frac{\delta}{2}\right),$$

$$\|\widetilde{\theta}_{t+1} - \theta^\star\|_{V_t} \leq \|\widetilde{\theta}_{t+1} - \theta^\star\|_{V_{t+1}} \leq \beta_{t+1}^{\theta^\star}\left(\frac{\delta}{2}\right) + \alpha_{t+1}\left(\frac{\delta}{2}\right)$$

By the triangle inequality of norm $\|\cdot\|_{V_t}$, it holds that

$$\|\widetilde{\theta}_t - \widetilde{\theta}_{t+1}\|_{V_t} \leq \|\widetilde{\theta}_t - \theta^\star\|_{V_t} + \|\widetilde{\theta}_{t+1} - \theta^\star\|_{V_t}$$
$$\leq 2\beta_{t+1}^{\theta^\star}\left(\frac{\delta}{2}\right) + 2\alpha_{t+1}\left(\frac{\delta}{2}\right),$$

where the last inequality holds due to the monotonicity in $\{\beta_t^{\theta^\star}\left(\frac{\delta}{2}\right)\}_{t=1}^T$ and $\{\alpha_t\left(\frac{\delta}{2}\right)\}_{t=1}^T$.

Therefore, we have built up the confidence ellipsoid for $\widetilde{\theta}_t - \widetilde{\theta}_{t+1}$ as

$$\|\widetilde{\theta}_t - \widetilde{\theta}_{t+1}\|_{V_t} \leq 2\beta_{t+1}^{\theta^\star}\left(\frac{\delta}{2}\right) + 2\alpha_{t+1}\left(\frac{\delta}{2}\right),$$

which fits into the condition of Lemma A.10 and leads to the result that

$$\sum_{t=1}^{T-1} \sum_{i=1}^{M} \left| \langle \widetilde{\theta}_t - \widetilde{\theta}_{t+1}, x_{t,i} \rangle \right| \leq \mathrm{RGT}\left(2\beta_T^{\theta^\star}\left(\frac{\delta}{2}\right) + 2\alpha_T\left(\frac{\delta}{2}\right)\right).$$

$\square$

## D.10   Proof of Claim A.15

*Proof.*   •  $\mathbb{E}_{\theta^\star \sim \pi}\left[B_f^{\theta^\star}\right]$.

Recall that $\theta^\star$ is coming from the prior $\mathcal{N}\left(\mathbf{0}, \lambda^{-1}\mathbf{I}\right)$, so $\theta^{\star(i)} \overset{\text{i.i.d.}}{\sim} \mathcal{N}\left(0, \lambda^{-\frac{1}{2}}\right)$, then

$$\mathbb{E}\left[\|\theta^\star\|\right] \leq \sqrt{\mathbb{E}\left[\|\theta^\star\|^2\right]} = \sqrt{\frac{d}{\lambda}},$$

$$\mathbb{E}\left[B_f^{\theta^\star}\right] = 2L \cdot \mathbb{E}\left[\|\theta^\star\|\right] = O\left(\sqrt{\frac{d}{\lambda}}L\right).$$

- $\mathbb{E}_{\theta^\star \sim \pi}\left[\mathrm{RGT}\left(\beta_T^{\theta^\star}\left(\frac{1}{2T}\right) + \alpha_T\left(\frac{1}{2T}\right)\right)\right].$

  Recall from (A.16)the definition of $\mathrm{RGT}\left(\eta_T(\delta)\right)$ as

  $$\mathrm{RGT}\left(\eta_T(\delta)\right) = \eta_T(\delta)\sqrt{\frac{2L^2 + 2\lambda}{\lambda}} \cdot \sqrt{dMT \log\left(\frac{\sigma^2 d\lambda + MTL^2}{\sigma^2 d\lambda}\right)} + \eta_T(\delta)\frac{2L}{\sqrt{\lambda}} \cdot dM \log\left(\frac{\sigma^2 d\lambda + MTL^2}{\sigma^2 d\lambda}\right)),$$

  in which only term $\eta_T(\delta)$ is $\theta^\star$ dependent.

  Also recall the definitions of $\beta_T^{\theta^\star}(\delta)$ and $\alpha_T(\delta)$ from (A.14) and (A.15), we have

  $$\mathbb{E}_{\theta^\star \sim \pi}\left[\beta_T^{\theta^\star}\left(\frac{1}{2T}\right)\right] \le \sqrt{2\log(2T) + d\log\left(\frac{\sigma^2\lambda d + TML^2}{\sigma^2\lambda d}\right)} + \sqrt{d}, \qquad \text{(D.23)}$$

  $$\mathbb{E}_{\theta^\star \sim \pi}\left[\alpha_T\left(\frac{1}{2T}\right)\right] = 2\sqrt{2d\log(2T)} + \sqrt{d}. \qquad \text{(D.24)}$$

  Plugging into $\eta_T(\delta) = \beta_T^{\theta^\star}(\delta) + \alpha_T(\delta)$, then

  $$\begin{aligned}
  \mathbb{E}_{\theta^\star \sim \pi}\left[\eta_T\left(\frac{1}{2T}\right)\right] &= \mathbb{E}_{\theta^\star \sim \pi}\left[\beta_T^{\theta^\star}\left(\frac{1}{2T}\right)\right] + \mathbb{E}_{\theta^\star \sim \pi}\left[\alpha_T\left(\frac{1}{2T}\right)\right] \\
  &\le \sqrt{2\log(2T) + d\log\left(\frac{\sigma^2\lambda d + TML^2}{\sigma^2\lambda d}\right)} + \sqrt{d} + 2\sqrt{2d\log(2T)} + \sqrt{d} \\
  &= O\left(\sqrt{d\log\left(\frac{\sigma^2\lambda d + TML^2}{\sigma^2\lambda d}\right)}\right). \qquad \text{(D.25)}
  \end{aligned}$$

  Therefore, we bound the order of $\mathbb{E}_{\theta^\star \sim \pi}\left[\mathrm{RGT}\left(\beta_T^{\theta^\star}\left(\frac{1}{2T}\right) + \alpha_T\left(\frac{1}{2T}\right)\right)\right]$ by

  $$\begin{aligned}
  &\mathbb{E}_{\theta^\star \sim \pi}\left[\mathrm{RGT}\left(\beta_T^{\theta^\star}\left(\frac{1}{2T}\right) + \alpha_T\left(\frac{1}{2T}\right)\right)\right] \\
  =&O\left(\frac{L}{\sqrt{\lambda}}d\sqrt{MT}\log\left(\frac{\sigma^2\lambda d + TML^2}{\sigma^2\lambda d}\right)\right) + O\left(\frac{L}{\sqrt{\lambda}}d^{\frac{3}{2}}M\log\left(\frac{\sigma^2\lambda d + TML^2}{\sigma^2\lambda d}\right)\right).
  \end{aligned}$$

  $\square$