# OpenReview forum: "Bandit Theory and Thompson Sampling-Guided Directed Evolution for Sequence Optimization"
_NeurIPS.cc/2022/Conference — NeurIPS 2022 Accept_

### Official Review · Reviewer_WM9h · 2022-07-06

**Rating:** 7
**Confidence:** 3
**Soundness:** 3 good
**Presentation:** 4 excellent
**Contribution:** 2 fair

**Summary:**

The paper introduces a variant of a linear Bandits model for directed evolution (DE). DE is concerned with iteratively optimizing a population of individuals by selecting a subset of promising individuals for mutation and recombination in each step. The utility of individuals is modeled with a linear function parameterized by a parameter theta. Theta is refined via a new variant of Thompson sampling. The difference to classic Thompson sampling is that the one cannot directly sample individuals due to the stochasticity of mutation and recombination. The method has sublinear regret bounds, that were confirmed in a simulation experiment. It was successfully applied to the optimization of CRISPR sequences.

**Questions:**

* In the Simulation Experiment in Section 6.1: How was the utility f chosen?
* If I understood correctly, the main difference to existing linear Bandits is that one cannot choose actions freely from the full action set. How does this differ from Sleeping Bandits setting?

**Limitations:**

The authors do not address the potential negative societal impact of their work. Their main application -- gene editing -- is an exemplary case for an ethically controversial topic. I see that the entire debate on this issue cannot be addressed in a single ML paper, but one could have at least pointed out that it is an issue and referred to more detailed discussions of it. In particular, because such cases are specifically mentioned in the submission guidelines.

**Strengths And Weaknesses:**

The paper is very well written. Even with no background on biotechnologies, one can read it in one go. The idea to include mutation and recombination into linear Bandits is new, as far as I know. The paper only mentions protein design optimization and gene editing, so I believe that its impact is mostly limited to biotechnology. I like that the paper not only claims that the method has real-world applications, but has already been able to show that the method has been successful in real-world applications. The results in the simulation experiment look promising too, however the baseline (the basic DE approach) seems to be rather simple. The theoretical claims are well supported.

---

> ### Author Response · Authors · 2022-08-02
> **Initial Response to Reviewer WM9h**
>
> Thanks for your helpful input on our paper. To address your questions and concerns:
>
> >In the Simulation Experiment in Section 6.1: How was the utility $f$ chosen?
>
> The utility $f$ was chosen to be linear with weights sampled from a multivariate Gaussian distribution, which exactly followed the assumption of f in the theory section. And we approximated the bayesian regret by averaging the regrets under multiple instances of $f$. We try to give our theoretical results justification via simulation to the most extent by doing so.
>
> >If I understood correctly, the main difference to existing linear Bandits is that one cannot choose actions free from the full action set. How does this differ from Sleeping Bandits' setting?
>
> Yes, technically the main constraint of our problem compared with simple linear bandits is not all actions are available to pick in each round, which makes our problem resembles the sleeping bandits. However, in each round, the available action set is generated with an underlying evolutionary nature, which limits one to a neighborhood that is reachable by mutation and recombination around the previous (group of) action(s) picked.
>
>
> >The results in the simulation experiment look promising too, however, the baseline (the basic DE approach) seems to be rather simple.
>
> We did new simulations to include a new baseline (LS) which uses linear regression to predict $f$ as a simple way to reuse information.  Please check the updated results in [Figure 1](https://imgur.com/a/TP1kG7k). (We refer you to our [response to all reviewers](https://openreview.net/forum?id=drVX99PekKf&noteId=sGmf7afovGZ) on the very top for more details about experimental setups.)
>
> The results demonstrate the faster convergence and advantage of TS-DE, especially when evaluations are subjected to a relatively high noise level. This is because Thompson sampling encourages exploration, which is necessary for a sublinear regret and fast convergence to the optimal solution under uncertainty.

---

> ### Author Response · Authors · 2022-08-04
> **Societal impact of gene-editing**
>
> Hello,
>
> We appreciate that you raise the concern about gene-editing and potential societal impact. This is a great and important question, and we would like to respond formally. The following discussion may be slightly beyond the scope of our bandit theory paper, but we are glad that you raised this question.
>
> In our companion paper, we designed the guide RNA sequence for a gene-editing tool that inserts a barcode into cells for "cell tagging". This technology is used for high-throughput omics sequencing, which allows scientists to track single-cell states. Our optimized sequence allows one to track cell states at a resolution 10+ folds higher than before. Note that this gene-editing tool is used completely **ex vivo, outside any living body**. This tool has nothing to do with editing animals or humans. **It applies to drug target discovery and enables high-throughput drug candidate screening, and we hope it will yield positive societal impacts**.
>
> We hope this can alleviate your concern about the gene-editing impact. Thank you very much!

---

> ### Author Response · Authors · 2022-08-09
> **follow up?**
>
> Dear Reviewer,
>
> Have you seen our response? Please let us know if you have any further question.
>
> As discussed below, our proposed method applied to a biotech for high-through omics and drug discovery.  The social impact will be positive. Given that this was the only concern you worried about, would you consider raising the score if it is no longer a concern?
>
> Thank you

---

> > ### Comment · Reviewer_WM9h · 2022-08-09
> > **Updated score**
> >
> >
> > I don't have the background knowledge to assess the societal impact, but if the authors have thought about it and address it, I am willing to believe it. I also appreciate the updated results with stronger baselines. I therefore raised my score.

---

> > > ### Author Response · Authors · 2022-08-09
> > > **thank you**
> > >
> > > Hi Reviewer, thank you for the update and for the discussions!

---

### Official Review · Reviewer_qppa · 2022-07-10

**Rating:** 6
**Confidence:** 2
**Soundness:** 3 good
**Presentation:** 3 good
**Contribution:** 3 good

**Summary:**

This paper study a cross domain problem for biological sequence optimization with bandit problem, and tries to provide theoretical understanding of directed evolution under bandit theory with Thompson sampling.

**Questions:**

stated in weaknesses.

**Limitations:**

empty

**Strengths And Weaknesses:**

# strengths
1. The problem is interesting.
2. The paper gives a theoretical understanding of bandit problem with directed Evolution.

# weaknesses
1. The linear setting is always a problem in real world application. I am not sure whether the linear bandit assumption suitable enough for real application.
2. I do not go through the proof, while it may seems that the proof may directly follow the standard proof sketch of bandit theory. It will be much better if the authors provide a brief description about the proof difficulties during adopting the  typical TS bandit proof technique.

---

> ### Author Response · Authors · 2022-08-02
> **Initial Response to Reviewer qppa**
>
> Thank you for reviewing our paper! Please note that we address some of the common questions in the general response above. To address your other questions:
>
> >It will be much better if the authors provide a brief description of the proof difficulties during adopting the typical TS bandit proof technique.
>
> Basically, the difficulty lies in the constraint that one is limited by the actions that are reachable by mutation and recombination operators from the current population in the DE process. It means that optimal action under each function estimate $f_t$ may not be available, so the score of $S_t$ under $f_t$ is less than the optimal value of $f_t$, which we refer to as an optimization gap. Due to this constraint, the total regret of DE is greater than that of linear bandit by a series of accumulated optimization gaps, which however is not too large, fortunately, and proving this is the most challenging part of the analysis.
> Thanks for your suggestion, we will add a short description to the latest version.

---

> > ### Author Response · Authors · 2022-08-09
> > **follow up?**
> >
> > Dear Reviewer,
> >
> > This is a gentle reminder to check if you have read our response and have any follow up question?
> >
> > In case you might have missed to see our common response to all reviewers about the linear model assumption. You can find it here: https://openreview.net/forum?id=drVX99PekKf&noteId=sGmf7afovGZ
> > In short, the linear model was commonly used by biologists for sequence-to-function prediction, and its analysis turns out to be highly nontrivial due to the recombination step. Two other reviewers have acknowledged that their concerns are addressed and have raised the scores.
> >
> > Please let us know if we have addressed your concern and if you have further suggestions. Thank you very much!

---

### Official Review · Reviewer_fbLu · 2022-07-10

**Rating:** 5
**Confidence:** 5
**Soundness:** 2 fair
**Presentation:** 2 fair
**Contribution:** 2 fair

**Summary:**

This paper focuses on a specific application **local** directed evolution. It starts with a set of candidate sequences and proposes an algorithm that performs two operations on the batch of sequences: recombination and common point mutation. The response function is modelled as a linear function of d protein motifs, where each motif can be or off. Authors analyze Bayesian regret of their procedure supported with comparison to classical evolutionary strategy.

**Questions:**

- Module 1, Line 6, x' should be z, right?
- How do you select the population of sequences S? Do you apply any diversity there?
- What is basic DE in your opinion?
- What is this modulus of contraction?
- Figure 6.1, what is the message the graph implies when we consider \mu?
- Can you explain 6.2 left better, I did not get it.


**Limitations:**

- The paper tries to give a very general introduction to directed evolution, however in 4 projects involving DE evolution never was the goal to change motifs of the protein and work in this greedy fashion. Instead enzymatic proportions were to be improved using a few selected mutations in the vicinity of the active site. I think the formulation authors have is fine and is probably motivated by their experimental setup, but this is **a form** of directed evolution not **the directed evolution**. I want to stress that this is an important point. If the goal of this paper is to introduce this problem to the broader ML community, we better do this carefully without over generalizing. I am fine with the setup, but one has to clearly say that this is a specific setup of DE. For example, in my opinion, this paper does not address  the major challenges in DE e.g. epistasis, combinatorial problems etc.
- Lacking theoretical understanding. I am not sure the proof is correct, since there is a magic quantify appearing suddenly. This is my justification for a lower score. I think the theorem might be correct, but the way there is not clearly stated. Also, I see no reason why scaling in d^2 is necessary for such a simple objective (linear on a hypercube).
- The paper lacks an algorithmically meaningful baseline, which could be executed synthetically.



**Strengths And Weaknesses:**

Strengths
----------
- I like the formalization attempt of the experimental pipeline authors are using and designing an algorithm for it
- Its nice this found a real-world application and improved over classical evolutionary approaches

Weakness
---------
- The authors do not do justice to the field of directed evolution. There are evolutionary based methods which operate on a current batch of sequences and amend it as described here - more aligned with local (or evolutionary approaches). However, there are other approaches which can synthesise specific mutants and operate on the combinatorial space likewise with high-throughput methods. Perhaps an important citation from this line of work is the paper which for  the first time uses BO with Gaussian processes and has regret guarantee which appeared in PNAS 2013, by authors Romero, Krause and **Arnold**. This is a ripe field that falls under the directed evolution keyword as well.

- Focus on regret minimization is a big minus. I see no reason why to focus on regret minimization. A more reasonable benchmark is to report the best variant so far. Thompson sampling often samples very greedy steps in order to achieve low regret, but in practice we do not want this at all, we want to explore given the uncertainty of $\theta$. We want to be as diverse as possible to find meaningful new information and candidates.

- There are two sub-methods introduced, first the operators are introduced in one order, explanation of them is in reversed order and they appear in the algorithm yet reversed.

- I have significant doubts about the proof of Theorem 5.2
  - namely the sudden emergence of modulus of concentration? Which I have no clue about.
  - the term H2, should be bounded by d\sqrt{TM} not the other exponent as authors propose
  - Also one hast to explicitly state that this is for linear objective supported on unit
hypercube. This proof does not work generally. A reader might be tempted to generalize to general linear bandits in general, but this is not true and it's not sufficiently clearly spelled out. However, due to the simplicity of the objective (hypercube + linear) I have no doubts that the algorithm would eventually converge since there is a random component due to $\mu$. (One just needs to learn d components sufficiently fast)

- The function in this case is the additive function of motifs, this is an extremely simplistic assumption for many practical applications where complex multi mutation epistasis occurs and is of interest to be modelled by data-driven methods.
- There isn’t any meaningful baseline algorithm for example and algorithm that would try to estimate the effect of all pointwise mutations, i.e. S-> S_1, first mutation, S->S_2 second mutation etc. This would probably converge in O(d\sqrt{TM}) steps as well and I believe even faster.

---

> ### Author Response · Authors · 2022-08-04
> **Initial Response to Reviewer fbLu (part 1 out of 2)**
>
> Thank you for reviewing our paper, and for sharing these suggestions!
>
> >**Question 1** The authors do not do justice to the field of directed evolution. There are evolutionary-based methods that operate on a current batch of sequences and amend it as described here. However, there are other approaches that can synthesize specific mutants and operate on the combinatorial space likewise to high-throughput methods. Perhaps an important citation from this line of work is the paper [1] which for the first time uses BO with Gaussian processes and has a regret guarantee which appeared in PNAS 2013.
>
> Thank you for the comments. We feel that the blame “the authors do not do justice to the field of directed evolution” is quite harsh. We have been collaborating with a bio expert on directed evolution, and have consulted for expert opinions many times when writing the paper.
>
> We strongly agree that **our formulation is a theoretical simplification and does not apply to every practice of directed evolution**. Our goal is to analyze a mathematical model for an evolving (aka, via mutation and recombination) process. We’d be glad to discuss more relevant approaches, add missing citations from the field of biology, and acknowledge that our proposed model is substantially simplified.
>
> In addition, we are not sure if paper [1] mentioned by the reviewer counts as directed evolution. It studied a new biotechnology that “can synthesize specific variants and operate on the combinatorial space likewise with high-throughput methods” .  To our best understanding, this type of approach seems to explore the landscape in a more aggressive manner and thus does not have a typical evolutionary (mutation+recombination) nature.  In the abstract of [1], it says that the new method “ … **contrasts with directed evolution**...”, which seems to suggest that it is a different method. We are not experts in biology, and feel that this discussion is really beyond the scope of our paper.
>
> We definitely will add more discussions and make sure to give a fair overview of this biotechnology field, and we really appreciate your guidance. However, a fully thorough discussion about “what counts as directed evolution” is a subject in biology, and may be beyond the scope of machine learning.
>
> [1]: Romero, Philip A., Andreas Krause, and Frances H. Arnold. "Navigating the protein fitness landscape with Gaussian processes." Proceedings of the National Academy of Sciences 110.3 (2013): E193-E201.
>
> >**Question 2** Focus on regret minimization is a big minus. I see no reason why to focus on regret minimization. A more reasonable benchmark is to report the best variant so far.
>
> Intuitively, DE is guiding a population of variants moving towards the optima using surrogate functions, which is naturally a low regret process. We think focusing on regret minimization takes advantage of this evolutionary nature and gives a population-based result.
> Note that the **quality of the best variant found so far can be easily derived from the regret bound**
> $$\hbox{Optimality Gap of Best Variant So Far} \leq \frac1{TM} Regret(T)$$
> A small regret immediately implies the population is dominated by optimal variants, with a provable error bound.
>
> >**Question 3** I have significant doubts about the proof of Theorem 5.2
>
> **We are confident that our proof is correct**. The full proof (in the supplementary) is quite lengthy and technical. We understand that it may take multiple hours to fully digest. We hope to alleviate your doubts about the proof:
>
> - >namely the sudden emergence of modulus of contraction? Which I have no clue about.
>
> We guess your confusion comes from the gap going from the ascent property (5.2) (rewritten under function estimate $F_t$ and for population $S_{t-1}$ and $S_t$) :
>
> $\mathbb{E}\left[ F(S^{\prime}) \right] \geq F(S) + \frac{1}{\sqrt{2d}} \sum_i \left| \theta_i \right| \textrm{Var}_i(S)$, (1)
>
> to a contraction in the suboptimality gap of $S_t$ compared to that of $S_{t-1}$:
>
> $\mathbb{E} \left[ F^*_{t} - F_{t}(S_{t}) \right] \leq \gamma(F^*_{t} - F_{t}(S_{t-1}))$,  (2)
>
> where $\gamma$ shows up.
> To put it simply, the reason why (1) yields (2) is that the progress term
>  Associated with $S_{t-1}$ is proportional to ($(1-\gamma)$ times) its suboptimality gap  towards $F_t$*
> when $S_{t-1}$ is not too close to a corner of the hypercube, which is ensured by the perturbation that mutation introduces. Please refer to **Lemma A.7** and its proof in appendix A for a formal and detailed derivation.
>
> - >Term H2, should be bounded by $d\sqrt{TM}$ not the other exponent as the authors propose
>
> Indeed a term like H2 is usually bounded by $d\sqrt{TM}$ in bandit literature, where feature vectors belong to a unit ball. How problem is different: each sequence (action) is associated with a feature with the $l_2$ norm of $\sqrt{d}$ instead of 1. Consequently, the regret bound should be inflated by a factor of $\sqrt{d}$, and our result is correct.

---

> > ### Author Response · Authors · 2022-08-04
> > **Initial Response to Reviewer fbLu (part 2 out of 2)**
> >
> > - >Also one has to explicitly state that this is for linear objective supported on the unit hypercube. This proof does not work generally. Due to the simplicity of the objective (hypercube + linear), I have no doubts that the algorithm would eventually converge since there is a random component due to $\mu$.
> >
> > In section 3.2, Definition 3.1 and Assumption 3.2 explicitly stated the objective is linear and we consider binary feature embedding.
> >
> > **Convergence of evolutionary algorithms, even for optimizing a linear function given its $0$-th order oracle on the Boolean space {$0,1$}^d, is highly nontrivial.** Analysis is challenging due to the algorithms’ evolutionary nature. In fact, there was a series of papers in theoretical computer science studying the convergence of evolution algorithms on an even simpler problem known as ONEMAX [13, 22, 52, 37] (cited in our submission): optimizing over {$0,1$}^d given the $0$-th oracle of $f = \sum_{i}^{d} x_i$. Most of these papers studied evolutionary algorithms including only mutation operation but no recombination. See Section 2 in our paper for related works.
> >
> > Further, our paper studied the regret of evolutionary algorithms in the bandit setting, where only noisy partial information of the model is available so that estimating the model (exploitation) and exploration are required. It makes the regret analysis even more challenging than the convergence analysis.
> >
> > >**Question 5**: There isn’t any meaningful baseline algorithm.
> >
> > We did new simulations to include a new baseline (LS) which uses linear regression to predict $f$ as a simple way to reuse information.  Please check the updated results in [Figure 1](https://imgur.com/a/TP1kG7k). (We refer you to our [response to all reviewers](https://openreview.net/forum?id=drVX99PekKf&noteId=sGmf7afovGZ) on the very top for more details about experimental setups.)
> >
> > The results demonstrate the faster convergence and advantage of TS-DE, especially when evaluations are subjected to a relatively high noise level. This is because Thompson sampling encourages exploration, which is necessary for a sublinear regret and fast convergence to the optimal solution under uncertainty.
> >
> > >**Question 6**: Figure 6.1, what is the message the graph implies when we consider $\mu$?
> >
> > Given a sequence $x$, a common mutation is to mutate every site $x_i$ on the sequence independently (w.p. $\mu$, $x_i$ is resampled, otherwise, $x_i$ keeps the same), we call it uniform mutation (uniform on each site). However, if deploying the uniform mutation with a constant $\mu$, the evolution process will never converge to the optimal. To make it converge, a scheduling of $\mu$ is required (e.g. $\mu_t = 1/t$), which usually needs to be tweaked for good performance.
> >
> > Monitored by the function estimate, TS-DE only mutates the sites where a sequence has a low score (referred to as directed mutation). This design makes mutation adaptive to the evolution process: at an initial stage, the function estimate is not stable due to the lack of data, then mutations are frequently activated, visiting sequences in different directions. While as the function estimate stabilizes, mutations are less frequently activated until shut down.
> >
> > Since the mutation always helps the sequence improve its score under the current f estimate, theoretically, the higher the mutation rate, the faster TS-DE converges, which is also justified by the simulation results shown in Figure 6.1 (right).
> >
> > **Other questions**:
> > - >Module 1, Line 6, $x^{\prime}$ should be $z$, right?
> >
> > Yes, thank you for pointing it out. Typo corrected.
> >
> > - >How do you select the population of sequences S? Do you apply any diversity there?
> >
> > We select the variants purely based on their score under the current function estimate from a pool of variants generated after mutation and recombination; we don’t force diversity in selection.
> >
> > - >What is basic DE?
> >
> > See the footnote on page 8: “The basic DE approach does not employ any function estimate. It does random mutation with a predefined mutation rate and random crossover recombination. It evaluates every candidate sequence and uses the noisy feedback in replace of $f_{\tilde \theta}$ for selection.”
> >
> > - >Can you explain 6.2 left better, I did not get it.
> >
> > Sure, on the left of Figure 6.2, we visualized the evolution trajectory of $S_t, t =1, \cdots, T$ in one run of TS-DE. The blue contour in each subfigure is a projection of population $S_t$ ($M \times d $ matrix) onto the 2D plane, which is realized by first compressing the $M \times d $ matrix to $M \times 2$ via PCA and then using the KDE plot package to render the density contour.

---

> > ### Comment · Reviewer_fbLu · 2022-08-05
> > **discussion**
> >
> > Q1. Well I still strongly believe that one should not claim this is **the** directed evolution. The other approach is what is more modernly associated with application of ML to directed evolution. You should give more context to what this field is. You are bringing these ideas to ML community and it is great but you need to approach this holistically and not claim this is the solution to directed evolution otherwise you introduce confusion if other people call something directed evolution - they will be looking for similar operators as you do have. This is a large field with a lot of problem setups, and you have a specific random mutagenesis bio-setup, which is valid and fine, just not completely representative, and I advise to make the writing and introduction more holistic.
> >
> > Q2. There is a whole line of algorithm called *best-arm identification* which are much more suited for simple-regret or identifying the best mutant. Of course you can bound simple regret but average cumulative, but there is a huge gap which motivates those 20+ papers that you can find under that keyword. Your choice for cumulative regret is convenience, which is acceptable but should be mentioned as limitation.
> >
> > Q3. In went over A.7 and I am convinced now. What remains is "By a standard
> > argument of concentration and a union bound taken over i ∈ [d], with probability 1 − δ" in Lemma A.4, how do you get the high probability result i.e. log(1/\delta) type?
> >
> > Q4. Potential lemma, I see where you problem lies and you need to absorb the dimension due to the hypercube. You are right, usually you do this in two steps, first potential lemma then bound on the norm. This is what confused me. I do not know whether you should state it as suboptimality as you do in the paper, and compare it one-to-one with existing literature. This is not directly comparable.
> >
> > In order to increase my score, I ask you to have a more holistic introduction, and **clearly** state you work on a hypercube.

---

> > > ### Author Response · Authors · 2022-08-06
> > > **Response to discussion**
> > >
> > > Hello, thank you for the prompt response, and for willing to increase the score!
> > >
> > > **I will increase my score if you give a more holistic introduction and state clearly you work on a hypercube**.
> > >
> > > Thank you for the suggestion. We didn’t mean to overstate the generality of our model, and we never implied that it should apply to the entire field of DE. Sorry if this might have caused any confusion, and we agree that this point should be remarked *clearly* in the paper.
> > >
> > > We drafted the following remark and added to the introduction. Please find it below and let us know if you have further suggestions and related papers that we should cite:
> > >
> > > > \paragraph{Important Remark}
> > >
> > > >The scope of this work is to provide a simplified mathematical model and basic theoretical understanding of an evolutionary-based process that is common in directed evolution. We emphasize that our framework is a theoretical simplification, assuming linear objective over a hypercube. In real-world experimental systems,  one needs to consider prior knowledge about the system and applying any machine learning method would require careful calibration and customization.
> > >
> > > >We acknowledge that protein engineering is a rich field and it is not restricted to methods that are based on mutagenesis and recombination. Our framework only applies to a specific type of evolution-based process, but there exists many other DE methods that are not evolution-based. Protein sequence engineering and directed evolution, more broadly speaking, constantly evolves as new biotechnologies keep emerging. For example, new biotechnology makes it possible to synthesize specific variants and operate on the combinatorial space likewise with high-throughput method, and this allows directly applying a Gaussian process bandit algorithm [1]. See [2] for a high-level survey of this active area of research, and see [3,4] for more examples. This active and exciting field brings many new opportunities for machine learning.
> > >
> > > > [1] Romero, Philip A., Andreas Krause, and Frances H. Arnold. “Navigating the protein fitness landscape with Gaussian processes.” Proceedings of the National Academy of Sciences 110.3 (2013): E193-E201.
> > > [2]Yang, Kevin K., Zachary Wu, and Frances H. Arnold. “Machine-learning-guided directed evolution for protein engineering.” Nature methods 16, no. 8 (2019): 687-694.
> > > [3] Fox, Richard J., S. Christopher Davis, Emily C. Mundorff, Lisa M. Newman, Vesna Gavrilovic, Steven K. Ma, Loleta M. Chung et al. “Improving catalytic function by ProSAR-driven enzyme evolution.” Nature biotechnology 25, no. 3 (2007): 338-344.
> > > [4]Bedbrook, Claire N., Kevin K. Yang, Austin J. Rice, Viviana Gradinaru, and Frances H. Arnold. “Machine learning to design integral membrane channelrhodopsins for efficient eukaryotic expression and plasma membrane localization.” PLoS computational biology 13, no. 10 (2017): e1005786.
> > >
> > >
> > > **Best-arm identification** Thank you for the suggestion. Indeed best-arm identification is very relevant and an important problem to study. Given that the protein sequence space is typically huge, looking for the exact best arm may be unrealistic given the limited experimental resource. How to mathematically model this question and designing provable algorithms would be a very interesting research direction for future work.
> > >
> > > **Lemma 4, how do you get the high probability result i.e. log(1/\delta) type**?
> > >
> > > In short, the proof works by applying concentration inequalities and a union bound. For the i-th feature dimension, we used a concentration inequality to show that a positive portion of the mutated sequences obtain the favorable feature, where the factor log(1/\delta) showed up in the batch size from applying the Hoeffding-Azuma inequality. Then we applied a union bound over all feature directions to make it hold for all dimensions, which added another factor of log d to the batch size.

---

### Official Review · Reviewer_X52n · 2022-07-11

**Rating:** 5
**Confidence:** 4
**Soundness:** 3 good
**Presentation:** 3 good
**Contribution:** 2 fair

**Summary:**

This paper proposes a novel approach to biological sequence optimization that models the task as a linear bandit problem and uses Thompson Sampling to guide a Directed Evolution algorithm towards optimizing the linear fitness function.  The approach ("TS-DE") is a variation of DE that is demonstrated to outperform classical (random/blind) DE alone in simulation that achieves a Bayesian regret that is optimal in population size and number of generations.
​
The DE and TS-DE algorithms are structurally the same, using crossover and point mutations to evolve a population of sequence variants over generations.  The difference is that TS-DE tracks a posterior of the fitness function between generations, and in each generation draws a fitness function estimate from this posterior, and (roughly speaking) limits the recombined variants and mutation positions to those which would see improvement under that fitness function estimate.  The result is more efficient convergence to optimal or near-optimal sequences.
​
It is not surprising on its own that a method which takes advantage of side information in crossover and mutation steps should outperform a completely random mutation approach.  It is disappointing that the authors did not compare this approach to at least one baseline that also also uses this side-information, such as training and scoring a linear model on proposed sequences to use as a filter between rounds.
​
Nonetheless, a key strength to the paper is its mathematical rigor, and the "main result" of the paper, that proves that TS-DE achieves a Bayesian Regret of $\tilde{O}(d^2\sqrt{MT})$.  The proof summary in the body of the paper was helpful for intuition, and great care was taken in the appendix to support each step of the proof.
​
The main limitation/weakness of both this main result and the algorithm itself arises from the two necessarily assumptions: (3.2) which requires the true fitness function to be a linear function with weights drawn independently from Gaussian distributions, and (3.5) which restricts measured values to having homoskedastic, iid, Gaussian, additive noise.  Neither of these assumptions are likely to hold, or even approximately hold, in practice, and there is no discussion of the sensitivity of the method and results to violations of these assumptions.
​
A secondary limitation/weakness is in the applicability of this method and results to biological experiments, for two main reasons: the linear model, and biological feasibility of intervening to filter between generations.  While the authors claim in Appendix 2 that the linear model can be generalized somewhat beyond binary motifs, it is not explained how this is done, and the application seems to still be restricted to linear models.  The biological feasibility / utility here may be a bigger issue, and is discussed in the Weaknesses section below.

**Questions:**

- What is the population size and selection strength for the DE and DE-TS approaches? It seems to me that a more realistic DE module would be more competitive with the DE-TS approach presented here, see section 5 in Sinai and Kelsic 2020 (https://arxiv.org/abs/2010.10614)

**Limitations:**

## Limitations
​
- The main limitations of this paper is it's reliance on strong and unlikely assumptions that the fitness landscape be linear and noise be iid Gaussian, and the questionable applicability in biological experiments identified in Weakness (7) - were not discussed in the paper.  Neither of these negate the contributions of the paper, but should be presented, and if possible, some discussion of the sensitivity of these results to violations in these assumptions should be added. I also think benchmarking against other sequence design work on more conventional challenges would strengthen the paper significantly. There is still a bit of a gap between the theoretical results and applicability that I'm not convinced this work would help close.
​
- It would be helpful if authors contrast their work with algorithms discussed by Sinai and Kelsic 2020 on model-guided sequence design, in particular discussion in section 5 (both with better DE modeling and contrasting with Algorithms like CbAS, Brookes et al 2019).
​
Minor
 Some grammatical issues:

1. In second paragraph "DE, one of the top molecular technology breakthrough in the past century, demonstrate human's ability to engineer proteins at will": breakthrough -> breakthroughs, demonstrate -> demonstrates, human's -> humanity's.

2. The following paragraph was especially confusing, since the topic sentence claimed that DE "remains expensive and time-consuming" but the subsequent sentences all defend the opposite claim that DE is "generally easy" and has been "exponentially improved."

3. There's also a small issue in Figure 1.1 in the recombination example, where a child with a dark gray motif 4 arises from parents without that motif.

4. The comment after Assumption 3.5, that "Our goal is to maximize the Bayesian regret" presumably was a mistake and the authors meant "to minimize" instead.
​
​

**Strengths And Weaknesses:**

## Strengths
​
1. The proposed Algorithm 1, to the reviewer's knowledge, is novel, though straightforward, and sufficient explanation has been provided for others to implement these ideas easily in simulation.
2. The paper's structure and organization was thoughtful and easy to follow, with key contributions highlighted and more technical steps relegated to the appendix.
3. The mathematical rigor is high, with great care taken in the appendix to explain both the intuition and logic behind all mathematical claims.  The approach to combining DE and Bandit Theory overall was creative and carefully developed.
4. The main result, that TS-DE achieves optimal Bayesian regret at least with respect to population size and number of generations, is strong.
5. Simulations show TS-DE outperforms DE handily - though this result is not surprising, it was worthwhile to confirm this through a demonstration.
​
​## Weaknesses
​
1. Several major assumptions would appear to not hold in practice.  Primarily, the assumptions that the fitness function is linear in a known, relatively small basis of binary motifs.  That aside, there is also the issue of accurately estimating the "known" variance sigma, the assumption that that noise is uniform (while, in practice, heteroskedasticity is common).  The paper does not address the sensitivity of the performance of their approach or its bounds under minor violations of these assumptions.support on real data
​
2. An initial concern was that the linear motif assumption in the paper was not easily generalizable to realistic protein engineering settings.  In Appendix B the authors explain that, in real world applications, these assumptions were loosened, so that individual base pairs were considered for mutation, and features were scalar-valued (not binary-valued).  While more details on these extensions would have been appreciated, it is understandable that they may be saved for the other publication mentioned and/or remain proprietary.

3. One way to alleviate these concerns is to actually show the performance of the algorithm on some simple but more realistic tasks, e.g. this package allows the authors to test their algorithms against some competing methods (https://github.com/samsinai/FLEXS).
​
4. Unclear how the mutation rate mu is chosen.  Presumably the choice of mu has some impact on the convergence of the algorithm if not bounds, yet this parameter does not appear in the analysis or experiments.
​
5. Novelty of the approach - while this exact formulation is novel, there are other recent examples in the literature that combine Thompson Sampling with Bandit problems, such as https://arxiv.org/pdf/2205.10113.pdf.  The method in this paper seems to be very similar to the paper under review, though it frames itself as a multi-armed bandit approach, and therefore compares itself against multi-armed bandit algorithms, including UCB methods.  Moreover, while the Thompson Sampling element of the algorithm is key to proving the main result (the bound on regret), in effect the algorithm (Alg 1) itself comes out as a fairly standard directed evolution approach
​
6. The paper claims that the resulting regret bound is optimal in M and T, and this is remarked upon, but not proven or cited.
​
7. A key argument for the linear bandit model is made in the Remark at the end of Section 2.  In short, it says that this problem is NOT a multi-armed bandit problem, primarily because we are not free to choose any action, but are limited by biology to mutation and recombination.  This is the reason why this method is not compared against multi-arm bandit methods in simulation, and why its regret is not compared to the regret of multi-arm bandits.  At first this seems entirely reasonable.  However, Algorithm 1 introduces crossover and mutation steps that involve sequence filtering based on calculating model scores, and intervening at this step.  This adds hugely significant cost to DE in a biological setting.  In classic/random DE, the crossover and mutation steps are random, not because of a lack of side-knowledge that could in theory be used to direct these steps, but because the biological steps of mutation and crossover naturally occur randomly.  While it is trivial "in silico" to intervene and filter out undesirable mutants according to side information, doing so in practice would require sequencing after evolutionary rounds to see what was produced, and then somehow separating out desirable and undesirable variants in the population before continuing.  Alliteratively, the whole mutation process could be done in silico, and then the populations $S_t$ could be constructed by hand from the in silico list of variants, but doing this completely negates the advantage of DE as an inexpensive process AND negatives the Remark at the end of Section 2 explaining why the model is limited to linear bandits.  If we are going to allow this level of filtering and/or individual construction of variants, we shouldn't have to limit ourselves to variants that could potentially arise from DE mutation and crossover steps.  Given the description in Appendix 2 of applying this model in practice, and the use of CRISPR technology in its implementation, it seems likely that the sequences were synthesized from an in silico selection step that could have easily been expanded to include sequences unobtainable through crossover or limited point mutations, again raising the question of whether the the assumption that "this is not a multi-armed bandit problem" is *really* a limitation of the biology or just a desirable limitation for the sake of the mathematical results.

---

> ### Author Response · Authors · 2022-08-04
> **Initial Response to Reviewer X52n (part 1 out of 2)**
>
> Thank you very much for your suggestions! Your review is very thorough and gives several useful pointers. We benefit from these suggestions and really appreciate them :)
>
> >**Linear model assumption** The main limitation/weakness of both this main result and the algorithm itself arises from the two necessarily assumptions: (3.2) which requires the true fitness function to be a linear function (3.5) which restricts measured values to having homoskedastic, iid, Gaussian, additive noise.
>
> Linear models (and their infinite-dimensional version, kernel models) are commonly used in protein engineering, for example [1, 2, 3] as well as in our companion paper on CRISPR gene-editing. To our best understanding, although simple, it is one of the most popular models that biologists use in practice.
>
> Further, note that evolutionary methods were never studied before in bandit theory. Thus we feel that considering a linear model as well as i.i.d. Gaussian noise is quite standard and a necessary first step. More complicated nonlinear models and dealing with heteroskedastic noise will be interesting directions of future work.
>
> [1] Chase R Freschlin, Sarah A Fahlberg, and Philip A Romero. Machine learning to navigate fitness landscapes for protein engineering. Current Opinion in Biotechnology, 75:102713, 2022.
>
> [2] Zaugg, J., Gumulya, Y., Malde, A. K. & Bodén, M. Learning epistatic interactions from sequence-activity data to predict enantioselectivity. J. Comput. Aided Mol. Des. 31, 1085–1096 (2017).
>
> [3] Yang, Kevin K., Zachary Wu, and Frances H. Arnold. "Machine-learning-guided directed evolution for protein engineering." Nature methods 16, no. 8 (2019): 687-694.
>
>
>
> >**Applicability to biological experiments.** (1)While it is trivial "in silico" to intervene and filter variants, doing so in a wet lab would add up the extra cost due to sequencing and separating variants in order to utilize the side information. (2)Also, if the whole evolution process could be done in silico and then construct the final population by hand from an in-silico list of variants, then there is no reason to limit ourselves to strictly following an evolutionary fashion.
>
> Thank you for raising this point! We agree that our model is a theoretical simplification of an evolutionary method, and we don’t expect it to apply to every practice in biological experiments. It is possible that some experiment systems don’t limit themselves to strictly follow an evolutionary fashion, which will bring new opportunities for ML.
>
> We share your concern that using machine learning to successfully lower the cost of experiments requires more consideration and prior knowledge. Actually, this concern was also mentioned in the survey paper [1]: “Machine learning is not necessarily useful for all protein engineering applications. In cases where the screen (evaluation) is expensive or slow enough to outweigh the cost and time of sequencing and synthesis, machine learning is beneficial. The decision to use machine learning (and to use which machine learning method) should consider prior knowledge about the system (the difficulty of the screen, the smoothness of the fitness landscape, etc.)”. For a practical experiment system, whether or not one should intervene/synthesize variants “in silico” is a question beyond the scope of our ML theory paper.
>
> [1] Yang, Kevin K., Zachary Wu, and Frances H. Arnold. "Machine-learning-guided directed evolution for protein engineering." Nature methods 16, no. 8 (2019): 687-694.
>
> >**Baseline** It is disappointing that the authors did not compare this approach to at least one baseline that also uses this side-information, such as training and scoring a linear model on proposed sequences to use as a filter between rounds.
>
> We did new simulations to include a new baseline (LS) which uses linear regression to predict $f$ as a simple way to reuse information.  Please check the updated results in [Figure 1](https://imgur.com/a/TP1kG7k). (We refer you to our [response to all reviewers](https://openreview.net/forum?id=drVX99PekKf&noteId=sGmf7afovGZ) on the very top for more details about experimental setups.)
>
> The results demonstrate the faster convergence and advantage of TS-DE, especially when evaluations are subjected to a relatively high noise level. This is because Thompson sampling encourages exploration, which is necessary for a sublinear regret and fast convergence to the optimal solution under uncertainty.

---

> > ### Author Response · Authors · 2022-08-04
> > **Initial Response to Reviewer X52n (part 2 out of 2)**
> >
> >  >**Recent seemingly similar results** such as [this work](https://arxiv.org/pdf/2205.10113.pdf), whose method seems to be very similar to the paper under review.
> >
> > Thanks for bringing up this paper to our attention! We will cite it and discuss it properly. After going through this paper, we found it is solving a rather different problem. It seems to be an empirical method for solving a standard multi-arm bandit problem by evolving a group of posteriors, whose parameters are mutated and evolved, and without giving theoretical bounds. In contrast, our problem is about evolving a set of sequences for sequence optimization and we focus on regret bounds. Their approach doesn’t apply to our problem of directed evolution and doesn’t affect our novelty. But we agree that the connection is quite interesting and will discuss it!
> >
> > > **It would be helpful if authors contrast their work with other sequence design algorithms, e.g. AdaLead, DbAS and CbAS.**
> >
> > Thank you for the pointer! Both the [package](https://arxiv.org/pdf/2010.02141.pdf) [1] and the [primer](https://arxiv.org/abs/2010.10614) [2] are wonderful resources for anyone who is interested in implementing or developing sequence design methods.
> >
> > We checked out these algorithms. DbAS/CbAS (a robust version of DcAS) [3, 4] is based on a generative model and allows querying variants that are not reachable by using DE. AdaLead uses a nearest neighbor approach for prediction while our model assumes a linear objective. Due to the different settings, It wouldn’t be fair to compare these methods with DE numerically. Also they don’t have theoretical results to compare with.
> >
> > [1] Sinai, Sam, Richard Wang, Alexander Whatley, Stewart Slocum, Elina Locane, and Eric D. Kelsic. "AdaLead: A simple and robust adaptive greedy search algorithm for sequence design." arXiv preprint arXiv:2010.02141 (2020).
> >
> > [2] Sinai, Sam, and Eric D. Kelsic. "A primer on model-guided exploration of fitness landscapes for biological sequence design." arXiv preprint arXiv:2010.10614 (2020).
> >
> > [3] Brookes, David H., and Jennifer Listgarten. "Design by adaptive sampling." arXiv preprint arXiv:1810.03714 (2018).
> >
> > [4] Brookes, David, Hahnbeom Park, and Jennifer Listgarten. "Conditioning by adaptive sampling for robust design." In International conference on machine learning, pp. 773-782. PMLR, 2019.
> >
> >
> > >**Mutation rate** Presumably the choice of $\mu$ has some impact on the convergence of the algorithm if not bounds, yet this parameter does not appear in the analysis or the experiments.​
> >
> > See Theorem 5.2, the total regret is bounded by $\tilde O \left( \frac{d}{\mu \sqrt{\lambda}} \cdot d \sqrt{MT} \right)$ when mutation rate $\mu$ and batch size $M$ satisfying the condition $M = \Omega \left(  \frac{\log(d T)}{\mu^2} \right)$. In the experiment, we varied the value of $\mu$ and compared the results in Figure 6.1 (Right).
> >
> > >**Optimality of the final bound** The paper claims that the resulting regret bound is optimal in $M$ and $T$, and this is remarked upon, but not proven or cited.​
> >
> > The following papers [20], [29] were cited inside “Remark on regret bound”. They contained the regret lower bounds that matches with ours (in terms of $T$ and $M$, up to polylog terms).
> >
> > [20]Yanjun Han, Zhengqing Zhou, Zhengyuan Zhou, Jose Blanchet, Peter W Glynn, and Yinyu Ye. Sequential batch learning in finite-action linear contextual bandits. arXiv preprint arXiv:2004.06321, 2020.
> >
> > [29]Cem Kalkanlı and Ayfer Ozgur. An improved regret bound for thompson sampling in the gaussian linear bandit setting. In 2020 IEEE International Symposium on Information Theory (ISIT), pages 2783–2788. IEEE, 2020.
> >
> >
> > >**Grammatical issues**
> >
> > Thanks for pointing them out, we already revised the current version to fix these issues.

---

> ### Author Response · Authors · 2022-08-09
> **follow up?**
>
> Dear Reviewer,
>
> Have you seen our response? We have been discussing with other reviewers. Two of them have acknowledged that their concerns are addressed and have raised the score.  So we look forward to hearing from you. Please let us know if you have any further question, and we will make sure to address them all.
>
> Thank you!

---

> > ### Author Response · Authors · 2022-08-09
> > **new remark about applicability**
> >
> > Dear Reviewer,
> >
> > We have been discussing with another reviewer about applicability to bio experiment and the gap between theory and practice. Following you and the other reviewer's suggestion, we drafted a remark  in the introduction section to acknowledge the gap. See it below and please let us know if you have further suggestions and know about related work that we should cite.
> >
> >
> > > \paragraph{Important Remark}
> >
> > >The scope of this work is to provide a simplified mathematical model and basic theoretical understanding of an evolutionary-based process that is common in directed evolution. We emphasize that our framework is a theoretical simplification, assuming linear objective over a hypercube. In real-world experimental systems,  one needs to consider prior knowledge about the system and applying any machine learning method would require careful calibration and customization.
> >
> > >We acknowledge that protein engineering is a rich field and it is not restricted to methods that are based on mutagenesis and recombination. Our framework only applies to a specific type of evolution-based process, but there exists many other DE methods that are not evolution-based. Protein sequence engineering and directed evolution, more broadly speaking, constantly evolves as new biotechnologies keep emerging. For example, new biotechnology makes it possible to synthesize specific variants and operate on the combinatorial space likewise with high-throughput method, and this allows directly applying a Gaussian process bandit algorithm [1]. See [2] for a high-level survey of this active area of research, and see [3,4] for more examples. This active and exciting field brings many new opportunities for machine learning.
> >
> > > [1] Romero, Philip A., Andreas Krause, and Frances H. Arnold. “Navigating the protein fitness landscape with Gaussian processes.” Proceedings of the National Academy of Sciences 110.3 (2013): E193-E201.
> > [2]Yang, Kevin K., Zachary Wu, and Frances H. Arnold. “Machine-learning-guided directed evolution for protein engineering.” Nature methods 16, no. 8 (2019): 687-694.
> > [3] Fox, Richard J., S. Christopher Davis, Emily C. Mundorff, Lisa M. Newman, Vesna Gavrilovic, Steven K. Ma, Loleta M. Chung et al. “Improving catalytic function by ProSAR-driven enzyme evolution.” Nature biotechnology 25, no. 3 (2007): 338-344.
> > [4]Bedbrook, Claire N., Kevin K. Yang, Austin J. Rice, Viviana Gradinaru, and Frances H. Arnold. “Machine learning to design integral membrane channelrhodopsins for efficient eukaryotic expression and plasma membrane localization.” PLoS computational biology 13, no. 10 (2017): e1005786.

---

### Author Response · Authors · 2022-08-03
**Response to All Reviewers**

We thank all reviewers for their valuable feedback. We first respond to the shared questions raised from the reviews and then answer each reviewer’s questions separately.

>**Is the linear model assumption too strong?**

Linear models (and their infinite-dimensional version, kernel models) are commonly used in protein engineering, for example [1, 2, 3] as well as in our companion paper on CRISPR gene-editing. To our best understanding, although simple, it is one of the most popular models that biologists use in practice.

Further, note that evolutionary methods were never studied before in bandit theory. Thus we feel that considering a linear model is a necessary first step. More complicated nonlinear models will be interesting directions of future work.

[1] Chase R Freschlin, Sarah A Fahlberg, and Philip A Romero. Machine learning to navigate
fitness landscapes for protein engineering. Current Opinion in Biotechnology, 75:102713, 2022.

[2] Zaugg, J., Gumulya, Y., Malde, A. K. & Bodén, M. Learning epistatic interactions from sequence-activity data to predict enantioselectivity. J. Comput. Aided Mol. Des. 31, 1085–1096 (2017).

[3] Yang, Kevin K., Zachary Wu, and Frances H. Arnold. "Machine-learning-guided directed evolution for protein engineering." Nature methods 16, no. 8 (2019): 687-694.

>**Does linear model make analysis trivial?**

Convergence of evolutionary algorithms, even for optimizing a linear function given its $0$-th order oracle on the Boolean space {$0,1$}^d, is highly nontrivial. Analysis is challenging due to the algorithms’ evolutionary nature. In fact, there was a series of papers in theoretical computer science studying the convergence of evolution algorithms on an even simpler problem known as ONEMAX [13, 22, 52, 37] (cited in our submission). Most of these papers studied evolutionary algorithms including only mutation operation but no recombination. See Section 2 in our paper for related works.

Further, our paper studied the regret of evolutionary algorithms in the bandit setting, where only noisy partial information of the model is available so that estimating the model (exploitation) and exploration are required. It makes the regret analysis even more challenging than the convergence analysis.


>**New Baseline and Simulations**

Considering the suggestions on baseline to compare with TS-DE, we added a new baseline (LS) which uses linear regression to predict $f$ as a simple way to reuse information.  In the new simulation, we implemented LS combined with 3 different mutation strategies.
- LS_1/t: LS with uniform mutation (apply mutation on all dimensions independent) under rate scheduling $\mu_t = 1/t$
- LS_const.: LS with uniform mutation of constant rate $\mu = 0.5$.
- LS_0.5: LS with directed mutation (Module 2) of rate $\mu = 0.5$.

Please check the updated results in [**Figure 1**](https://imgur.com/a/TP1kG7k):
- (Left) Regret curves of LS_1/t and LS_const. are plotted to compare with basic DE and TS-DE as baseline, see Figure 1(left).
- (Right) Compared TS_DE ($\mu = 0.5$) with LS_0.5 in a noisier environment (setups : $d = 30$,  $M = 5$, noise level $\sigma = 10.0$).

The results demonstrate the faster convergence and advantage of TS-DE, especially when evaluations are subjected to a relatively high noise level. This is because Thompson sampling encourages exploration, which is necessary for a sublinear regret and fast convergence to the optimal solution under uncertainty.

---

### Meta-Review · Area_Chair_egQN · 2022-08-30

**Recommendation:** Accept
**Confidence:** Certain

**Metareview:**

The initial round of reviews for the submitted manuscript was mostly positive in tone, but this enthusiasm was tempered by a number of deep technical issues -- and some more philosophical issues regarding the presentation and framing of the results -- raised by the reviewers. Fortunately, the author rebuttal and author--reviewer discussion phases went a long way toward clearing up some initial confusion and clarifying the contributions of the authors, which swayed the prevailing opinion of the reviewers toward acceptance.

I want to commend the authors for their enlightening contributions to that discussion, which assuaged most of the reviewers' initial complaints.

However, I would also like to stress that it is critical that the fruits of this discussion (especially with reviewers X52n and fbLu) be incorporated into a revised version of this manuscript. The reviewers are unanimous in this opinion.

**Award:**

No

---

### Decision · Program_Chairs · 2022-09-14

Accept